# Involvement of condensin in cellular senescence through gene regulation and compartmental reorganization

Osamu Iwasaki [1,4], Hideki Tanizawa [1,4], Kyoung-Dong Kim[2], Andrew Kossenkov[3], Timothy Nacarelli[3], Sanki Tashiro[1], Sonali Majumdar[3], Louise C. Showe[3], Rugang Zhang [3] & Ken-ichi Noma [1*]

Senescence is induced by various stimuli such as oncogene expression and telomere shortening, referred to as oncogene-induced senescence (OIS) and replicative senescence (RS), respectively, and accompanied by global transcriptional alterations and 3D genome reorganization. Here, we demonstrate that the human condensin II complex participates in senescence via gene regulation and reorganization of euchromatic A and heterochromatic B compartments. Both OIS and RS are accompanied by A-to-B and B-to-A compartmental transitions, the latter of which occur more frequently and are undergone by 14% (430 Mb) of the human genome. Mechanistically, condensin is enriched in A compartments and implicated in B-to-A transitions. The full activation of senescence genes (SASP genes and p53 targets) requires condensin; its depletion impairs senescence markers. This study describes that condensin reinforces euchromatic A compartments and promotes B-to-A transitions, both of which are coupled to optimal expression of senescence genes, thereby allowing condensin to contribute to senescent processes.

[1] Institute of Molecular Biology, University of Oregon, Eugene, OR 97403, USA. [2] Department of Systems Biotechnology, Chung-Ang University, Anseong 17456, Republic of Korea. [3] The Wistar Institute, Philadelphia, PA 19104, USA. [4] These authors contributed equally: Osamu Iwasaki, Hideki Tanizawa. *email: noma@uoregon.edu

Cell biological and genomic studies have revealed that genome packaging is not accomplished simply by randomly positioning genetic material[1,2]. Rather, a hierarchy of organizing processes work to form genomic structures of various sizes. These structures range from enhancer-promoter contacts, to topologically associating domains (TADs), and much larger A and B compartments[3,4]. The multi-megabase A and B compartments correspond to euchromatic and heterochromatic domains, respectively[5]. The A and B compartments tend to be separated from each other, representing in vivo genome organization[6]. The several hundred-kilobase TADs are organized by the architectural proteins CTCF and cohesin[7–9]. Some studies predict that TADs and sub-TADs are the genome-organizing units that co-regulate gene expression and coordinate replication timing[10–12]. However, other work shows that acute cohesin depletion, which diminishes TADs, has only a modest effect on gene regulation, indicating that much remains to be learned about how the topological domain organization regulates transcription[13].

Senescence is a state of stable cell growth arrest caused by exhausting the capacity for DNA replication and the ability to respond to various stresses, including DNA damage and excessive growth stimulation by oncogenic Ras[14]. Two types of senescence are recognized, oncogene-induced senescence (OIS), which is induced by oncogene expression, and replicative senescence (RS), which is induced by telomere shortening. Senescent processes involve radical changes in gene expression, especially genes encoding the senescence-associated secretory phenotype (SASP) factors[15,16]. These include interleukins, chemokines, growth factors, and matrix metalloproteinases[17,18]. In addition, autophagy and lysosomal genes are upregulated in senescent cells. Here we refer to SASP factors and other genes upregulated during senescent processes collectively as senescence genes[15].

Senescent processes are also accompanied by significant alterations in the global 3D genome architecture, such as the formation of senescence-associated heterochromatic foci (SAHF) and the distension of centromeric satellites[19,20]. Genomic studies suggest that OIS impairs heterochromatic TADs (lamin-associated domains), while RS enhances genomic contacts within 2 Mb and promotes global chromosomal compaction[21,22]. Moreover, it has been shown that the frequency of A/B compartmental transitions is limited in the early stage of RS[23]. In the premature aging disease Hutchinson-Gilford progeria syndrome, late passage progeroid cells are devoid of the A/B compartments[24]. This evidence collectively implies that 3D genome reorganization occurs during senescence, although how specific genome architectures are reorganized in senescent cells and contribute to the gene regulation that is characteristic of the senescent state remains largely unclear.

It is well described that the short-range interactions among enhancers and promoters are orchestrated by the structural maintenance of chromosomes (SMC) complex cohesin[8,13]. Recent studies suggest that another SMC complex, condensin, also plays critical roles in the formation of topological domains and chromosomal territories, although the detailed mechanisms remain much less well understood[25–31]. In terms of the roles of condensin in transcriptional regulation, it has been shown that enhancer-promoter contacts are potentially mediated by condensin in mouse and human cells[32,33]. In Drosophila and Caenorhabditis elegance, condensin has also been implicated in transcriptional regulation[27,34]. In yeast cells, condensin is involved in clustering RNA polymerase III-transcribed genes such as tRNA and 5 S rRNA[29,35]. Yeast condensin is recruited to highly transcribed genes and reduces unwound DNA derived from transcription[36–38]. It has been shown that a majority of condensin-mutation effects on transcript levels are derived from mis-segregation of yeast chromosomes and exosome, leading to

defects in RNA decay[39], although condensin has also been proposed to play a direct role in transcriptional regulation in non-dividing quiescent cells[40]. Therefore, the transcriptional roles of condensin are preferably examined in non-dividing cells, i.e., senescent cells, as in this article.

Humans have two condensin complexes, I and II, the latter of which is known to have genome-organizing functions due to its nuclear localization during interphase[41,42]. Our previous study revealed that condensin II participates in oncogene-induced senescence, although how it coordinates the 3D genome reorganization with transcriptional regulation during senescent processes remains elusive[43]. This genomic study suggests that a major function of condensin is to reinforce the euchromatic A compartments. Condensin localizes to senescence genes, such as SASP and p53 target genes. The combined findings suggest that condensin positively contributes to senescent processes, via reorganizing the 3D genome, reinforcing euchromatic A compartments, and upregulating transcription of senescence genes.

## Results

**Condensin frequently localizes at promoters and enhancers.** Our previous work showed that condensin II participates in oncogene-induced senescence, but the detailed mechanisms of how condensin reorganized the 3D genome and/or altered transcription remained unclear. To mechanistically understand how condensin functions during senescent processes, we first examined the genome-wide distribution of condensin II in cells which had undergone oncogene-induced senescence (OIS). OIS can be induced in human lung fibroblast IMR90 cells by the expression of oncogenic RAS[43]. We first characterized condensin II-specific antibodies to be used for ChIP-seq analysis (Supplementary Fig. 1; Supplementary Notes). We then performed ChIP-seq to map chromosomal locations of the CAP-H2 condensin II subunit and found that global binding profiles, predicted from endogenous CAP-H2 and exogenous CAP-H2-FLAG ChIP-seq data, were generally conserved between OIS and growing cells; results from biological replicas were similar (Supplementary Fig. 2a). In addition, we performed ChIP-seq experiments for another condensin subunit, SMC2, and observed that CAP-H2 colocalizes with SMC2 in OIS cells, suggesting that these condensin subunits interact and bind to their target loci as the condensin II complex (Supplementary Fig 2a, b). CAP-H2 binding sites in OIS and growing cells were predicted as common sites shared by multiple ChIP-seq data (Supplementary Fig. 2c; Supplementary Notes). Although the general tendency of CAP-H2 binding sites was similar between OIS and growing cells, we also noticed some OIS-specific CAP-H2 peaks, where both CAP-H2 and CAP-H2-FLAG proteins were enriched at senescence genes such as SASP genes (IL1B and IL6) in OIS cells, but not in growing cells (Fig. 1a). CAP-H2 was also more enriched at a p53 target gene, p21, in OIS cells compared to growing cells. RNA polymerase II and, to a lesser extent, an SMC1 cohesin subunit were also detected at the senescence genes. CAP-H2 proteins were occasionally detected at potential enhancers marked by histone H3K27ac, e.g., downstream of IL1B (Fig. 1a; Supplementary Notes).

To obtain a global view of where condensin binds, we categorized CAP-H2 binding sites by type of genetic element and found that CAP-H2 preferentially distributed at active gene promoters and potential enhancers (Fig. 1b; Supplementary Fig. 2d). This preferential localization of CAP-H2 at active promoters and potential enhancers is quite different than the relative composition of those genetic elements in the human genome (control). Consistent with previous observations[32,44], cohesin (SMC1) was detected at active promoters and potential enhancers, and CTCF frequently localized at other regions such as

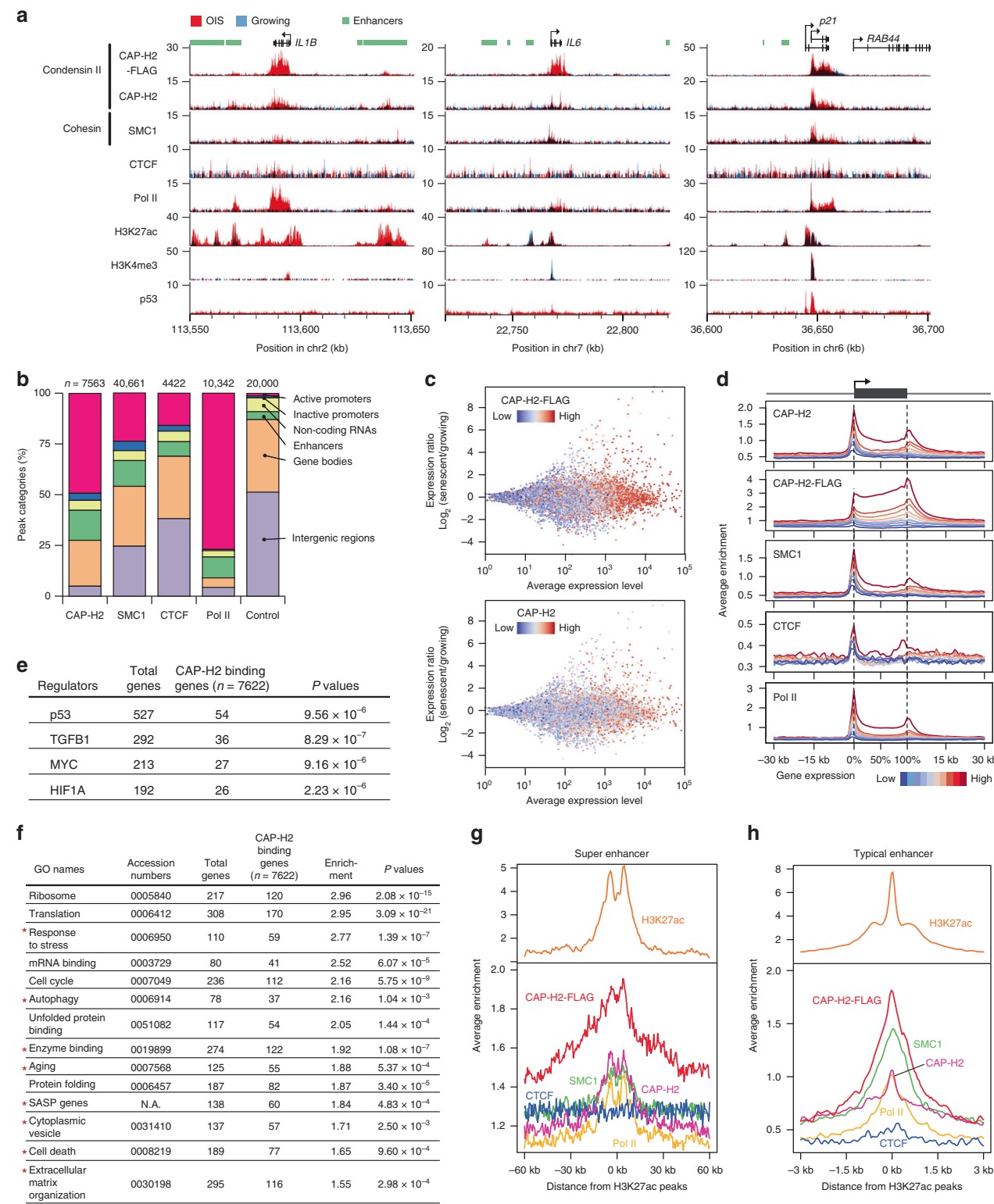

gene bodies and intergenic regions. As predicted, RNA polymerase II predominantly resided at active promoters. CAP-H2 peak numbers and genomic regions covered by CAP-H2 were expanded in OIS cells compared to growing cells (Supplementary Fig. 2e, f).

Testing for co-occupancy of genomic sites by CAP-H2 with the other genome-organizing factors and Pol II, we found that CAP-H2 significantly colocalized with SMC1 and Pol II ($P < 0.001$, hypergeometric distribution test; Supplementary Fig. 2g). The

genomic regions co-occupied by CAP-H2, SMC1, and Pol II were chiefly active promoters; regions shared by CAP-H2 and SMC1 were active promoters or potential enhancers (Supplementary Fig. 2h). By contrast, genomic sites occupied by both SMC1 and CTCF were frequently intergenic regions and gene bodies.

**Condensin localizes at senescence genes.** Since CAP-H2 often localizes at active promoters, we asked whether there was a

**Fig. 1 Condensin distributions in senescent (OIS) and growing cells. a** Distributions of condensin II subunit (CAP-H2 and CAP-H2-FLAG), SMC1 cohesin subunit, CTCF, and RNA polymerase II (Pol II) binding in OIS and growing IMR90 cells. Distributions of euchromatic histone H3K27ac and H3K4me3 marks and p53 were previously reported[47,68,69]. Positions of genes and enhancers are annotated on top. Enhancers are defined by H3K27ac peaks (Supplementary Notes). **b** Distributions of CAP-H2, SMC1, CTCF, and Pol II binding sites at the indicated genetic elements, in OIS cells. Numbers of total binding sites are shown at top. For the control, 20,000 loci were randomly selected from the entire genome and classified into the same categories. **c** Correlation between relative transcription levels (in OIS compared to growing cells) and CAP-H2 ChIP-seq enrichment. Expression ratios between OIS and growing cells (*Y*-axis) were plotted against average expression levels for respective genes (*X*-axis). Expression levels of genes were estimated based on RNA-seq data. Dot colors reflect ChIP-seq enrichment scores of CAP-H2-FLAG (top) and CAP-H2 (bottom) at respective genes. **d** Average ChIP-seq enrichment of the indicated proteins at Pol II-transcribed genes. Genes were classified into 10 groups based on their transcription levels in OIS cells. Each group contains 2,888 genes. Average binding patterns of the indicated proteins were plotted for the respective gene groups. Gene sizes from transcriptional initiation to termination sites were normalized to the same length for all the genes investigated. **e** Gene ontology (GO) analysis of CAP-H2 binding genes. CAP-H2 binding genes were cross-referenced with the genes controlled by key regulators. Significant regulators were defined by $P < 10^{-5}$ (Fisher exact test) and more than 20 CAP-H2 binding genes. **f** GO analysis for CAP-H2 binding genes to identify which genes involved in specific biological processes are preferentially bound by CAP-H2. Asterisks indicate groups that are significantly enriched with senescence upregulated genes (Fisher exact test). **g**, **h** Average enrichment of the indicated proteins and enhancer mark H3K27ac at super (**g**) and typical (**h**) enhancers in OIS cells. Potential super and typical enhancers are defined as described in Supplementary Notes. There were 1540 super and 39,531 typical enhancers predicted in OIS cells.

relationship between CAP-H2 enrichment and transcriptional levels. We performed RNA-seq analysis with OIS and control growing cells (see Methods) and found that transcription levels appeared to positively correlate with ChIP-seq enrichment of CAP-H2 condensin and SMC1 cohesin (Fig. 1c, d). Looking next at genes that were either significantly up- or downregulated in OIS cells compared to growing cells, we found that CAP-H2 enrichment was higher at significantly upregulated genes in OIS cells (Supplementary Fig. 2i; top panels in Supplementary Fig. 2j). Conversely, in growing cells, CAP-H2 enrichment was higher at significantly downregulated genes (bottom panels in Supplementary Fig. 2j), again suggesting that gene transcription levels positively correlate with CAP-H2 enrichment.

It has been shown that spurious ChIP-seq peaks are often derived from highly transcribed genes[45]. Therefore, we performed CAP-H2 ChIP-seq with OIS cells treated with RNA polymerase inhibitors (Triptolide and α-Amanitin). We also carried out nascent RNA visualization, RT-qPCR, and RNA-seq experiments, and confirmed that transcription was significantly impaired by Pol II inhibitor treatment (Supplementary Fig. 2k–m). We observed that CAP-H2 enrichment was retained at gene regions after the inhibitor treatment and that CAP-H2 enrichment was not correlated with nascent RNA levels (Supplementary Fig. 2n, o). These results collectively show that CAP-H2 tends to localize at highly transcribed genes and is retained at gene regions after the transcriptional reduction. Therefore, CAP-H2 peaks from ChIP-seq analysis likely reflect its endogenous distributions instead of spurious, non-specific peaks, although it is still unclear how the inhibitor treatment affects the protein accessibility at gene regions. It is also important to note that global gene expression levels were slightly higher in OIS cells than in growing cells ($P = 4.9 \times 10^{-9}$, two-sided Mann–Whitney *U* test; Supplementary Fig. 2k), suggesting that many genes are as actively transcribed in non-proliferating OIS cells as in proliferating cells.

Since condensin II localizes to active promoters and potential enhancers, it was possible that condensin preferentially localizes to specific groups of genes. To test this possibility, we performed gene ontology (GO) analysis for CAP-H2-bound active promoters, referred to here as CAP-H2 binding genes, and found that CAP-H2 binding genes were significantly enriched for the groups of genes controlled by particular regulatory factors, namely p53, TGFB1, MYC, and HIF1A (Fig. 1e). The GO analysis also revealed that CAP-H2 binding genes were significantly enriched for the groups of genes including SASP and other senescence genes and highly transcribed housekeeping genes, e.g., genes involved in ribosome biogenesis and translation (Fig. 1f). Consistent with the mapping data in Fig. 1a, endogenous

CAP-H2 and exogenous CAP-H2-FLAG were both highly enriched at super and typical enhancers (Fig. 1g, h). These results collectively demonstrate that condensin II localizes at senescence genes activated by the specific transcription regulators, highly transcribed housekeeping genes, and potential enhancers.

**Compartmental reorganizations upon OIS.** To begin studying the 3D genome reorganization during senescent processes, we applied the in situ Hi-C procedure[9] to OIS and growing IMR90 cells, and generated contact maps for every chromosome (Fig. 2a; Supplementary Fig. 3; Methods). The in situ Hi-C data were highly reproducible between biological replicates and correlated well with the standard Hi-C data (Supplementary Fig. 4a). To observe any global changes in chromatin contacts upon OIS, we then compared the contact probabilities between OIS and growing cells and observed that long-range contacts between heterochromatic domains, as marked by histone H3K9me3, were elevated in OIS cells; these contacts likely represent senescence-associated heterochromatic foci (SAHF) (Fig. 2b; Supplementary Fig. 3).

We next compared the general organization of the human genome into A and B compartments in OIS and growing cells (Fig. 2c–e; Supplementary Fig. 3; Supplementary Fig. 4b, c). The chromosomal locations of the A and B compartments were generally conserved between OIS and growing cells (Fig. 2c). B compartments were of similar size, typically 1–2 Mb (Fig. 2e; Supplementary Fig. 4c). However, A compartments in OIS cells were significantly enlarged (~50%) compared to A compartments in growing cells ($P = 5.55 \times 10^{-12}$, two-sided Mann–Whitney *U* test; Fig. 2e; Supplementary Fig. 4c). We also observed that heterochromatic B compartments in growing cells more frequently converted to euchromatic A compartments in OIS cells, termed BA transitions, compared with the opposite AB transitions (Fig. 2d, e; Supplementary Fig. 4b, c). This result was robust, as the BA transitions significantly overlapped between biological replicates (Bio 1 and Bio 2; $P = 1.42 \times 10^{-4257}$, hypergeometric distribution test; Supplementary Fig. 4d). The AB transitions also were conserved between biological replicates to some extent ($P = 1.50 \times 10^{-405}$, hypergeometric distribution test; Supplementary Fig. 4d). The genomic regions that underwent BA or AB transitions were usually small, i.e., 100–500 kb. It is noteworthy that approximately 14% (430 Mb) of the human genome underwent a BA transition upon OIS. The more frequent BA transitions than AB transitions likely resulted in the significant enlargement of A compartments in OIS cells compared to growing cells and in the consequent decrease in the number of A compartments upon OIS (Fig. 2e; Supplementary Fig. 4c).

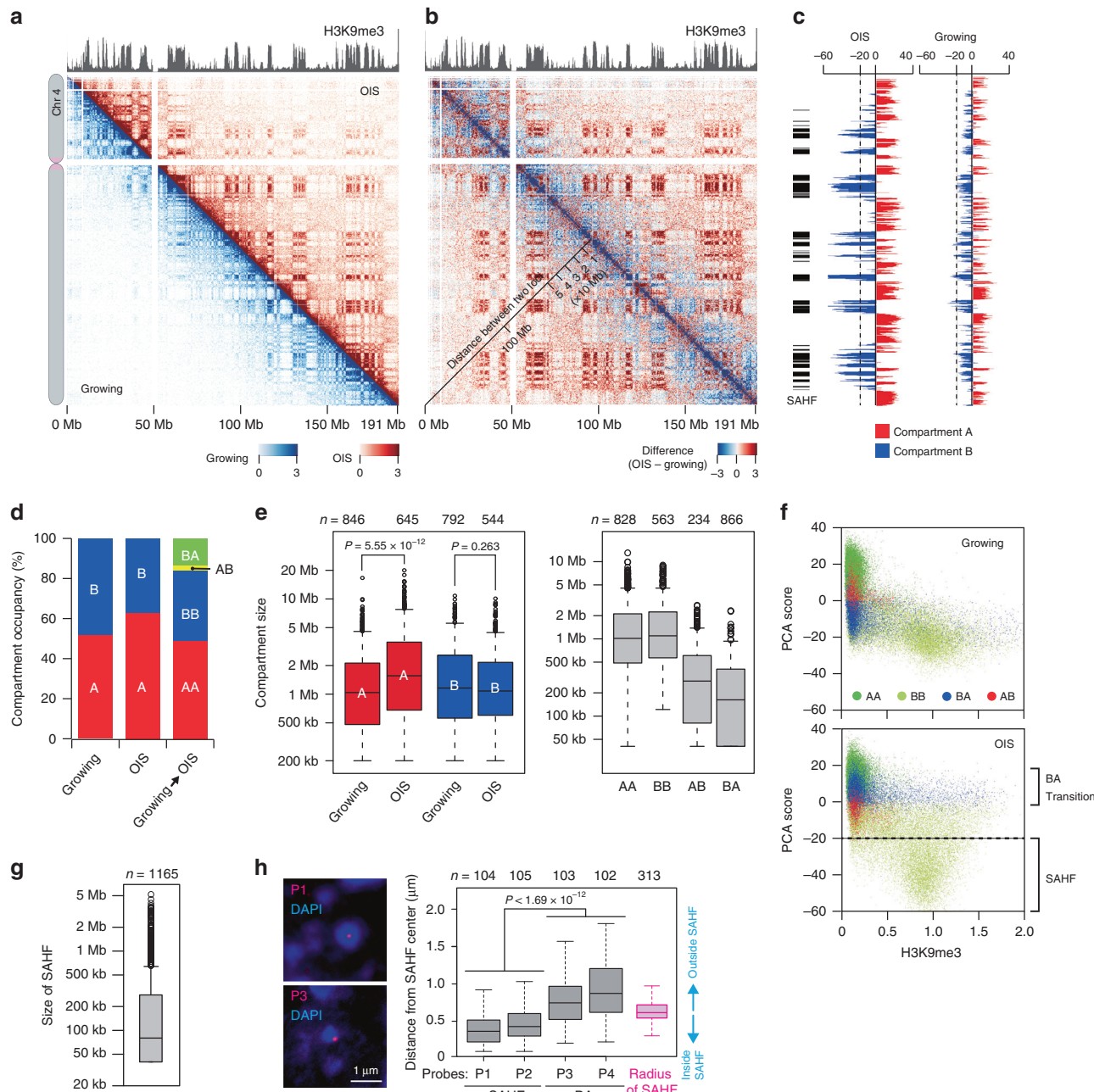

**Fig. 2 Compartmental reorganization and SAHF formation upon OIS. a** Contact maps for chromosome 4 in OIS (top right) and growing cells (bottom left) at 200 kb resolution. Contact maps were plotted as detailed in Supplementary Fig. 3. **b** Difference of contact probabilities between OIS and growing cells. Red and blue dots indicate that contact probabilities are higher in OIS and growing cells, respectively. **c** PCA (principal component analysis) scores in OIS and growing cells plotted along chromosome 4. PCA scores were calculated as described in Methods. SAHF were defined as genomic regions with PCA scores <−20; locations are indicated by black bars at left. **d** Occupancy of A and B compartments in growing and OIS cells (for replicate #1). Rightmost column shows compartmental switching (AB or BA) or not (AA and BB) between growing and OIS cells. Details are in Supplementary Fig. 4b. **e** Left: Size distributions of A and B compartments in growing and OIS cells (for the same data as in panel **d**; two-sided Mann–Whitney U test) shown as boxplots (central bar represents the median with boxes indicating the upper and lower quartiles, and whiskers extend to the data points, which are no more than 1.5× the interquartile range from the box; outliers shown as circles). Numbers of A and B compartments are shown at top. Right: Size distributions and numbers of genomic regions that belong to the respective compartmental categories. **f** Correlation between PCA score, H3K9me3 enrichment and chromatin compartment. PCA scores of every 40 kb bin across the human genome were plotted against histone H3K9me3 enrichment score in growing and OIS cells (for replicate #1). Dot colors represent genomic regions that belong to the indicated compartmental categories. **g** Size distribution of the SAHF defined in panel **f**. **h** FISH visualization of the SAHF-forming heterochromatic (P1 and P2 probes) and BA-switching loci (P3 and P4 probes) in OIS cells (Methods). Probe positions are shown in Supplementary Fig. 4h. Left: Examples of FISH images for the P1 and P3 probes. SAHF were visualized by DAPI staining. Right: The distance between a center of DAPI-dense area and FISH signal was measured in the indicated numbers of cells (top). Distance distributions shown as boxplots (central bar represents the median with boxes indicating the upper and lower quartiles, and whiskers extend to the data points, which are no more than 1.5× the interquartile range from the box) were compared between the SAHF and BA-switching loci (two-sided Mann–Whitney U test). DAPI signals were used to measure radiuses of SAHF.

Furthermore, frequent BA transitions upon OIS were also conserved in WI-38 cells[21], and BA and AB transitions upon OIS observed in IMR90 cells were significantly overlapped with those in WI-38 cells, implying that compartmental transitions upon OIS were partly conserved in different cell lines (Supplementary Fig. 4e, f). These results unexpectedly indicate that transcriptionally active or poised A compartments expand in non-propagating OIS cells compared to growing cells, which might explain why nascent transcripts are elevated in OIS cells compared to growing cells (Supplementary Fig. 2k).

**SAHF detection**. To further characterize the SAHF that were visualized in the difference map (Fig. 2b) and identify regions that likely formed the SAHF, we compared PCA scores and compartmental categories (i.e., AA, BB, BA, and AB) of every 40 kb bin across the genome. We found that genomic regions that maintained occupancy in a B compartment between growing and OIS cells, referred to as BB regions, were enriched by a histone H3K9me3 heterochromatic epigenetic mark and frequently had PCA scores below −20 in OIS cells but not in growing cells (Fig. 2f; Supplementary Fig. 4g).

We predicted that heterochromatic regions with PCA scores below −20 participate in SAHF formation (Fig. 2g). To verify our prediction, we performed FISH experiments to visualize the heterochromatic (P1 and P2 probes) and BA-switching loci (P3 and P4 probes) in OIS cells (Supplementary Fig. 4h). We observed that the heterochromatic loci were positioned significantly closer to the center of SAHF compared to the BA-switching loci ($P < 1.69 \times 10^{-12}$, two-sided Mann–Whitney $U$ test; Fig. 2h; Supplementary Fig. 4i), suggesting that heterochromatic loci with PCA scores below −20 are typically embedded within SAHF, whereas BA-switching loci predicted from the contact maps tend to reside at the euchromatic exterior of SAHF.

Since SAHF are formed by long-range contacts among heterochromatic regions, we asked whether the regions that had PCA scores <−20 were also regions that made long-range contacts. We calculated long versus short (LVS) scores, which represent the frequency of contacts between regions that are more than 2 Mb apart (Supplementary Fig. 4j). We found that genomic regions having LVSs scores >0.5 often coincided with genomic regions having PCA scores <−20 ($P < 0.001$, hypergeometric distribution test; Supplementary Fig. 4k). BB regions frequently had LVS scores >0.5 in OIS cells (Supplementary Fig. 4l), and AB and BB regions (i.e., B compartments in OIS cells) had higher LVS scores in OIS cells than in growing cells (Supplementary Fig. 4m). These results collectively suggest that heterochromatic B compartments tend to interact with one another via long-range contacts, resulting in the formation of SAHF in OIS cells.

**SAHF and transcriptional downregulation**. Our analysis predicted that SAHF were formed by long-range contacts among 1165 heterochromatic regions across the human genome (Fig. 2g). To seek a potential role of SAHF in transcriptional regulation, we performed gene set enrichment analysis (GSEA) using genes located within the 500–700 kb range from SAHF[46]. We found that this 200 kb window was significantly enriched with genes that were downregulated upon OIS (Fig. 3a). This 500–700 kb window was identified by examining regions located at specific distances from the SAHF (Fig. 3b–e), as follows. The <300 kb ranges did not show enrichment of downregulated genes, but the ranges from 0–400 kb to 0–600 kb were gradually enriched with downregulated genes, and the 0–700 kb and longer ranges became significantly enriched with downregulated genes (Fig. 3b). Breaking the distances into 200 kb windows, we found that regions within the 400–800 kb ranges from SAHF were

significantly enriched with downregulated genes (Fig. 3c). The percent of genes that were downregulated was highest in the 400–600 and 500–700 kb ranges from SAHF (Fig. 3d). Genes located within the 500–700 kb range from SAHF were more frequently downregulated compared to the control considering all human genes ($P = 0.00641$, two-sided Mann–Whitney $U$ test; Fig. 3e). These results together suggest that genes located within the 500–700 kb range from the SAHF-forming heterochromatic regions are occasionally and slightly downregulated upon OIS, potentially reflecting a weak position effect of SAHF organization on expression of genes in proximity.

Because longer chromosomes may contribute to SAHF formation more than shorter chromosomes, we compared the percent of the chromosome that was occupied by SAHF between longer (#1–14) and shorter (#15–22) chromosomes. We observed that SAHF occupied larger genomic regions in the longer chromosomes compared to the shorter ones (Fig. 3f). BA-switching regions showed the same tendency as SAHF, and the occupancy of SAHF and BA-switching regions was positively correlated, indicating that SAHF formation and BA transitions upon OIS tend to occur in the longer chromosomes (Fig. 3f, g).

**Condensin binding enhanced at B-to-A-switching regions**. To identify functional links between compartmental reorganizations and gene regulation upon OIS, we focused on the BA and AB transitions observed between growing and OIS cells (Fig. 2d). GSEA revealed that BA- and AB-switching regions were significantly enriched with up- and downregulated genes, respectively (Fig. 4a, b). We next confirmed that the euchromatic epigenetic marks (H3K4me3 and H3K36me3) were enriched at AA regions and that the heterochromatic mark (H3K9me3) was enriched at BB regions, by analyzing ChIP-seq enrichment of the histone marks in the respective compartmental categories (Fig. 4c; Supplementary Fig. 5a). Note that relative ChIP-seq enrichment of these epigenetic marks was not drastically altered between OIS and growing cells, for any compartmental categories[47] (Fig. 4c; Supplementary Fig. 5b). Quite differently, CAP-H2 condensin enrichment was altered upon OIS, being elevated and diminished at BA- and AB-switching regions, respectively, in OIS cells compared to growing cells (Fig. 4c; Supplementary Fig. 5b). SMC1 cohesin enrichment showed a similar tendency, although to a lesser extent. We also noticed that H3K27me3, a histone mark associated with inactive promoters, and LMNB1 (Lamin B1), a structural component of the nuclear lamina, were most enriched at BA-switching regions (Fig. 4c). These results collectively suggest that condensin binding to and dissociating from chromatin upon OIS potentially contributes to BA and AB transitions, respectively. Since ChIP-seq enrichment of the epigenetic marks at BA- and AB-switching regions was generally similar between OIS and growing cells (Fig. 4c; Supplementary Fig. 5b), it may be condensin binding and/or compartmental transitions that are responsible for the changes in expression of genes present at BA- and AB-switching regions.

To gain more detailed insights into how the larger A compartments observed in OIS cells are formed (Fig. 2e), we analyzed the genome-wide distributions of A and B compartments in OIS and growing cells and found that small B compartments were often converted to be parts of adjoining A compartments (Fig. 4d). This was consistent with the prior, genome-wide observation that BA transitions occurred more frequently than AB transitions, were small (100–500 kb), and likely caused the overall expansion in the size of A compartments in OIS cells (Fig. 2e; Supplementary Fig. 4c). Euchromatic H3K4me3 and heterochromatic H3K9me3 peaks were mainly positioned at AA and BB regions, respectively, as expected

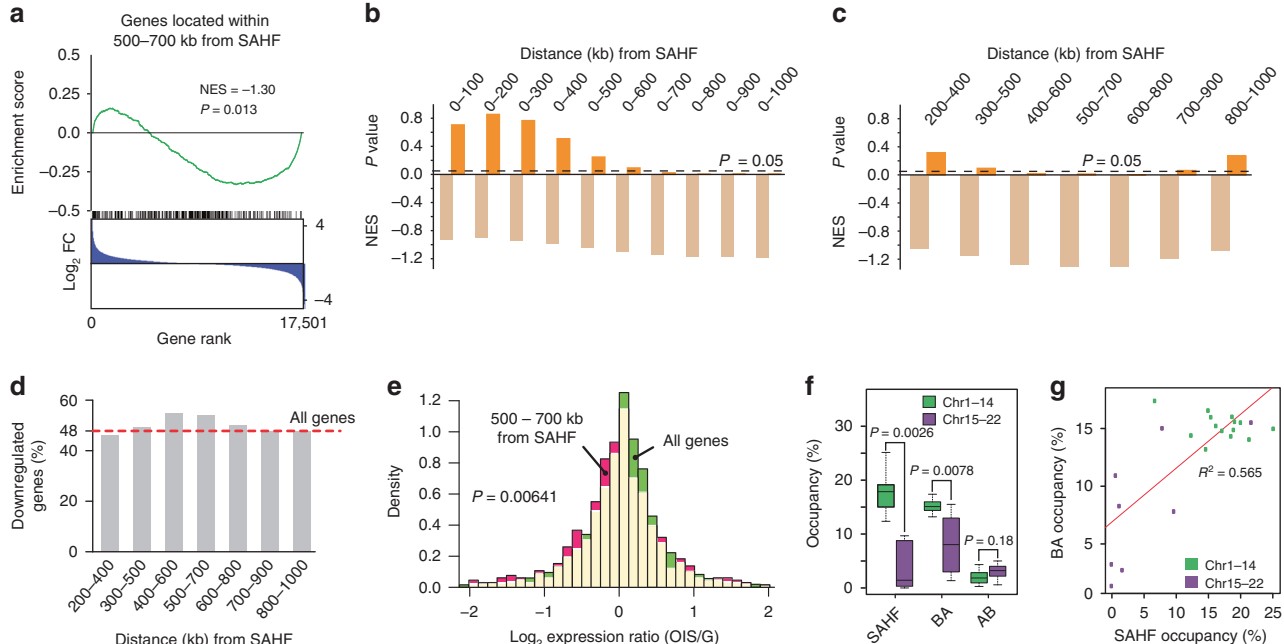

**Fig. 3 Transcriptional downregulation near SAHF-forming regions. a** GSEA of the 495 genes located within the 500–700 kb range from SAHF, from all chromosomes (Methods). All human genes were ranked by expression fold changes (FC), calculated as levels of gene expression in OIS cells divided by levels of expression in growing cells (blue distribution). The distribution of the target genes ($n = 495$) is shown by the row of black tick marks (also green line). The target genes were frequently downregulated in OIS cells compared to growing cells, indicated by the negative normalized enrichment score (NES). The significance of the biased distribution was assessed by an empirical phenotype-based permutation test[46]. **b** Statistical significance and NES of changes in gene expression, between OIS and growing cells, for groups of genes positioned within the indicated distances from SAHF, as analyzed using GSEA. **c** Same data as panel **b**, but GSEA was performed on groups of genes positioned within the indicated 200 kb windows from SAHF. **d** Percent of downregulated genes positioned within the indicated 200 kb windows. Note that 48% of all human genes were downregulated upon OIS, serving as a control distribution for downregulated genes. **e** Expression ratios between OIS and growing cells were calculated for the genes located within the 500–700 kb range from SAHF, and the distribution was plotted against the ratio. All human genes were also subjected to the same analysis (control). $P$-value was calculated by a two-sided Mann–Whitney $U$ test. **f** Percent of chromosome occupied by SAHF in chromosomes 1–14 and 15–22 shown as boxplots (central bar represents the median with boxes indicating the upper and lower quartiles, and whiskers extend to the data points, which are no more than 1.5× the interquartile range from the box). $P$-value was calculated by a two-sided Mann–Whitney $U$ test. BA- and AB-switching regions were analyzed identically. **g** Correlation between percent of chromosome occupied by SAHF and by BA-switching regions, in the indicated chromosome groups. The coefficient of determination ($R^2$) for a linear model fit (red line) was shown.

(Fig. 4d). Observations at the level of individual BA transitions revealed that CAP-H2 was more enriched at BA-switching regions in OIS cells compared to growing cells (Fig. 4e). Enrichment of the structural protein LMNB1 at the same region was comparable between cell types (Fig. 4e). In contrast, CAP-H2 was less enriched at AB-switching regions upon OIS (Supplementary Fig. 5c). GO analysis revealed that genes present at BA-switching regions were significantly enriched with the groups of genes including SASP and other senescence genes (Fig. 4f). These results indicate that condensin binding is positively correlated with BA transitions, where some senescence genes tend to be upregulated.

**Compartment borders often colocalize with TAD borders**. To seek structural links between the reorganization of topologically associating domains (TADs) and A/B compartments upon OIS, we predicted TADs in OIS and growing cells. Sizes and numbers of TADs were ~400 kb and 4300–5000, respectively, and comparable among all the in situ Hi-C samples (Supplementary Fig. 5d). TAD borders were more conserved among samples when border strength scores were higher (Supplementary Fig. 5e). Common TAD borders tended to be marked with CAP-H2 (condensin), SMC1 (cohesin), Pol II, H3K4me3, H3K36me3, but not H3K9me3 (Supplementary Fig. 5f).

Some TAD borders were present only in OIS or only in growing cells, referred to as OIS- and G-specific TAD borders, respectively. These cell type-specific borders were considerably less prevalent than 'non-specific' TAD borders, which were observed in both cell types (Fig. 5a). Interestingly, TAD borders that were present only in growing cells, G-specific TAD borders, often coincided with compartment borders that were present only in growing cells, G-specific compartment borders (a representative example in Fig. 5b). This result, that G-specific TAD borders were frequently positioned at G-specific compartment borders, was highly significant ($P < 0.001$, two-sided Mann–Whitney $U$ test; Fig. 5c). These results suggest that TAD borders demarcate A and B compartments in growing cells at least in some cases. Therefore, the disappearance of G-specific TAD borders upon OIS is coupled to the encroachment of A compartments and BA transitions (Fig. 5b).

**SAHF were not detected in RS cells**. We asked whether the SAHF formation we observed in OIS cells (IMR90 cell line) were conserved in RS, the form of senescence induced by telomere shortening. We applied the in situ Hi-C procedure to RS cells obtained from two different cell lines: IMR90 cells and foreskin fibroblast BJ cells (Fig. 6a, b). IMR90 and BJ RS cells were prepared as described in Methods; ~90% cells were positive for SA-β-gal staining, indicating that the majority were senescent

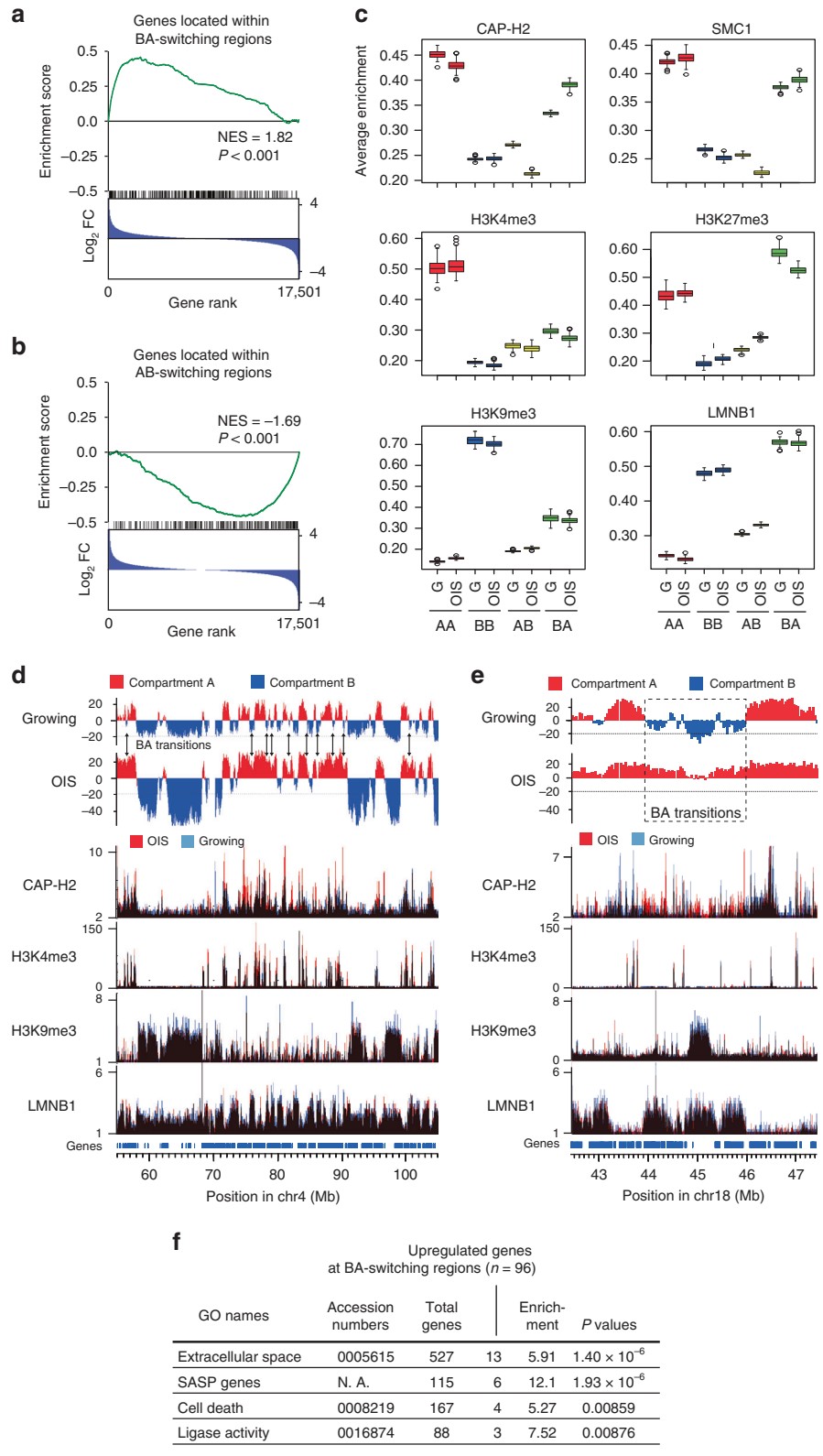

**f** Upregulated genes at BA-switching regions (*n* = 96)

| GO names | Accession numbers | Total genes | | Enrich-ment | *P* values |
|---|---|---|---|---|---|
| Extracellular space | 0005615 | 527 | 13 | 5.91 | $1.40 \times 10^{-6}$ |
| SASP genes | N. A. | 115 | 6 | 12.1 | $1.93 \times 10^{-6}$ |
| Cell death | 0008219 | 167 | 4 | 5.27 | 0.00859 |
| Ligase activity | 0016874 | 88 | 3 | 7.52 | 0.00876 |

(Supplementary Fig. 6a). DAPI foci were detected in about 20% of IMR90 RS cells but not in BJ RS cells (Supplementary Fig. 6b). This result is consistent with previous observations showing that the formation of DAPI foci in RS cells is dependent on p16 expression[48,49].

The in situ Hi-C data revealed that neither IMR90 nor BJ RS cells showed the global SAHF-derived contact pattern connecting heterochromatic domains observed in IMR90 OIS cells (Fig. 6a, b; Fig. 2a). Moreover, PCA scores in IMR90 and BJ RS cells were consistently above −20, in contrast to IMR90 OIS cells, which displayed PCA scores below −20 for BB regions (Supplementary Fig. 6c, d; Fig. 2f). This combined data suggests an absence of SAHF organization in RS cells. It was unclear why some microscopic images of IMR90 RS cells showed DAPI foci

**Fig. 4 Condensin is enriched at BA-switching regions in OIS cells. a** GSEA of genes located at BA-switching regions. All human genes were ranked by expression fold changes (FC), calculated as expression levels of genes in OIS cells divided by those in growing cells (blue distribution). For this GSEA, the difference in PCA scores between OIS and growing cells was calculated for each BA-switching region, and the top 300 genes present at the regions with the largest difference in PCA scores were analyzed. NES, normalized enrichment score. **b** GSEA of genes located at AB-switching regions ($n = 263$). **c** ChIP-seq enrichment of the indicated proteins and epigenetic marks for the respective compartmental categories in growing (G) and OIS cells. Five hundred genomic regions (40 kb bins) were randomly sampled from each compartmental category, and distributions of average ChIP enrichments are shown as boxplots (central bar represents the median with boxes indicating the upper and lower quartiles, and whiskers extend to the data points, which are no more than 1.5× the interquartile range from the box; outliers shown as circles). ChIP-seq data for the epigenetic marks and LMNB1 were previously published[47,70]. **d** An example of how BA transitions of small B compartments change the compartmental organization upon OIS. PCA scores in growing and OIS cells are plotted along a 50 Mb region in chromosome 4. BA transitions upon OIS are indicated by arrows. ChIP-seq data for CAP-H2, H3K4me3, H3K9me3, and LMNB1 are shown beneath. **e** Same as panel **d**, but a 5 Mb region with a single BA transition (dotted) is shown. **f** GO analysis of the 96 genes that are present at BA-switching regions and significantly upregulated upon OIS. These genes were cross-referenced with genes involved in specific biological processes to test the significance of their overlap (Fisher exact test).

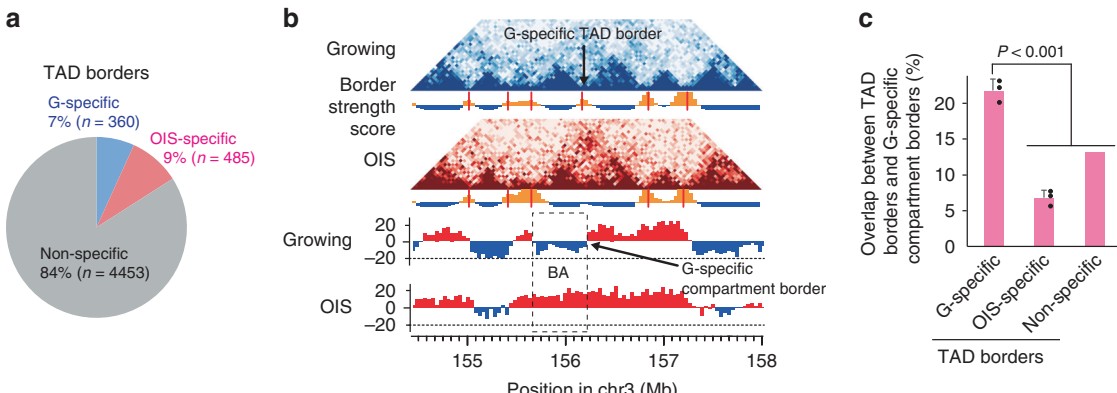

**Fig. 5 Colocalization of compartment and TAD borders in growing cells. a** Numbers of TAD borders that were specific to growing or OIS cells. TAD borders were predicted from the growing (Bio1 and Bio2) and OIS (Bio1, Bio2, and Bio3) data. TAD borders consistently present in growing cell data but not in OIS cell data were classified as G-specific borders; TAD borders conserved in at least two OIS replicates but not in the growing data were categorized as OIS-specific borders. Remaining TAD borders detected in at least two datasets were classified as non-specific borders. **b** An example showing the colocalization of a compartmental border that was only present in growing cells (i.e., G-specific compartment border) and a TAD border that was only present in growing cells (i.e., G-specific TAD border). Contact maps in growing and OIS cells are shown for the same 4 Mb region in chromosome 3 (top); contact maps are depicted as in Fig. 2a. TAD borders (narrow red vertical lines) were predicted from border strength scores and are shown beneath the maps (Methods). PCA scores in growing and OIS cells are shown at the bottom. **c** Frequency of overlap between G-specific TAD borders and G-specific compartment borders, as exemplified in panel **b**. Overlap between different types of TAD borders and G-specific compartment borders was counted. Positions of TAD borders were allowed to shift by ± 1 bin (40 kb) to estimate the overlap with G-specific compartment borders. *P*-values were calculated by two-sided Student's *t*-test, using biologically independent samples ($n = 3$, error bars represent the SD).

(Supplementary Fig. 6b), while the contact map for IMR90 RS cells (Fig. 6a) did not suggest the existence of SAHF. In this regard, we noticed that DAPI foci in IMR90 RS cells were slightly more diffuse compared to DAPI foci in OIS cells, and only 20% of IMR90 RS cells presented DAPI foci, potentially explaining the absence of a SAHF-derived contact pattern in IMR90 RS cells (Supplementary Fig. 6b).

**Compartmental reorganizations conserved between OIS and RS**. We next asked if common compartmental reorganizations occurred between OIS and the other form of senescence, RS. To do this, we first characterized RS-dependent compartmental reorganizations using the two different cell lines, IMR90 and BJ. We observed that upon RS the composition of A compartments was increased, in both IMR90 and BJ RS cells (by 1.98% and 5.94%, respectively; Supplementary Fig. 6e, f). The proportion of the human genome that underwent BA (or AB) transitions upon RS was similar for the two cell lines, accounting for approximately 8.30% (6.32%) and 10.88% (4.94%) of the genome, in IMR90 and BJ cells, respectively (Supplementary Fig. 6e, f). Genomic regions that underwent BA or AB transitions upon RS were usually 100–500 kb, similar between the cell lines, and similar to the size of transition regions observed upon OIS (Supplementary Fig. 6g, h;

Fig. 2e). Differently than for OIS cells, upon RS the sizes of both A and B compartments became significantly (~1.5×) enlarged, in the both cell lines; correspondingly, the numbers of A and B compartments decreased (Supplementary Fig. 6g, h). As the average size of A compartments increased by approximately the same amount (~1.5×) upon RS as upon OIS (Fig. 2e), the enlargement of A compartments may be a common senescent feature conserved between RS and OIS.

If the BA and AB transitions that occur upon OIS also occur upon RS, then this suggests that there are common compartmental reorganizations that occur among senesced cells, regardless of how the state of senescence was induced (i.e., by oncogene expression or telomere shortening). In the control analyses, where we assessed for overlap across replicates, we observed that BA transitions upon RS significantly overlapped between biological replicates, for the both cell lines ($P < 0.001$, hypergeometric distribution test; Supplementary Fig. 6i, j). AB transitions were also conserved between replicates, but to a lesser extent ($P < 0.001$, hypergeometric distribution test; Supplementary Fig. 6k, l).

To test for common reorganizations, we compared regions that undergo BA and AB transitions upon OIS and upon RS. Indeed, we observed significant overlap of BA transitions upon

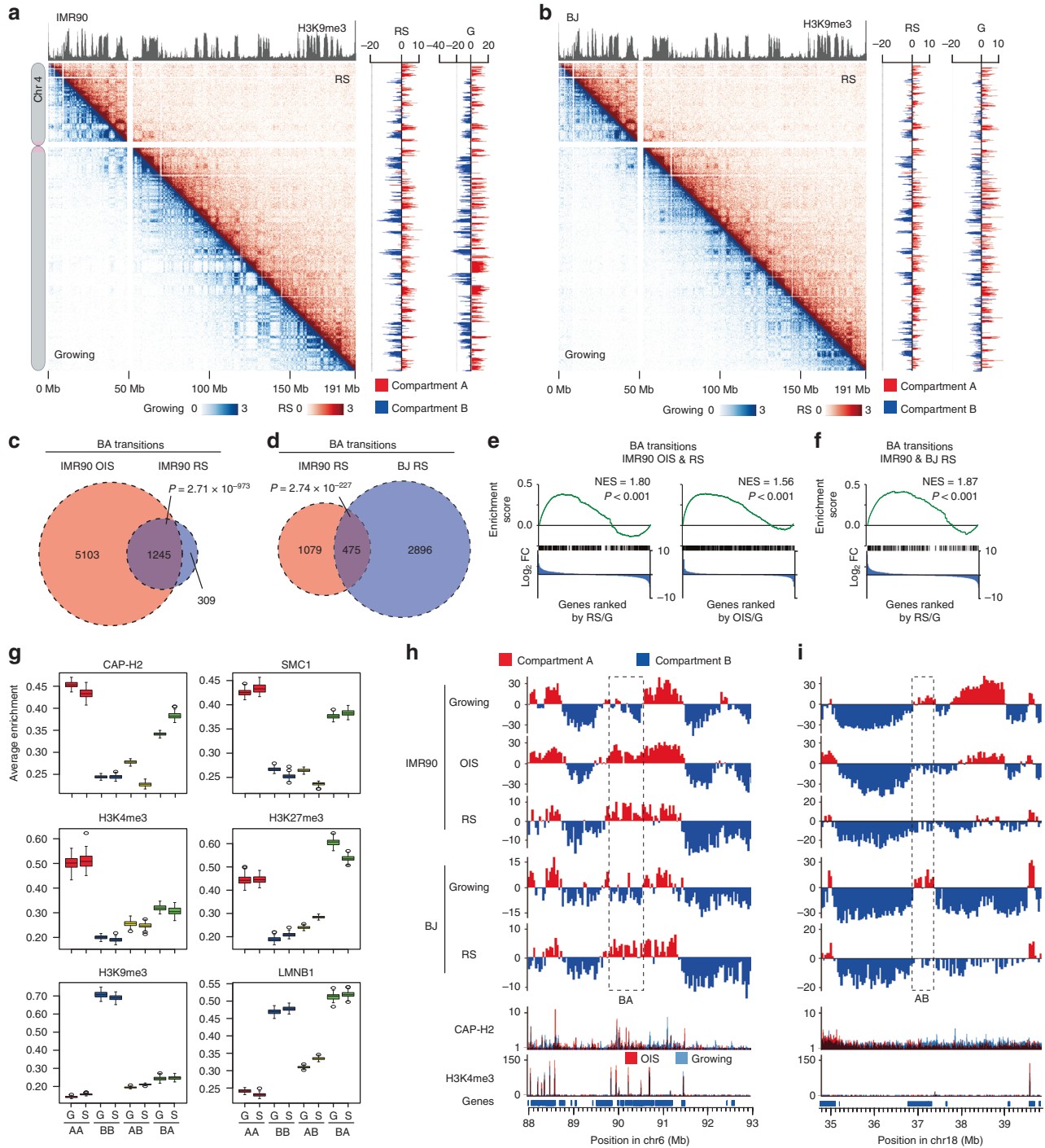

**Conserved compartmental transitions coupled to transcription.** To elucidate how compartmental reorganizations that are conserved between OIS and RS, here termed 'conserved reorganization regions', are connected to transcriptional regulation, we performed GSEA for genes present at the conserved reorganization regions. The conserved reorganization regions that underwent a BA transition were significantly enriched with upregulated genes (Fig. 6e, IMR90 cells). Such upregulated gene expression was also observed at BA-switching regions upon RS that were common between the two cell lines tested (Fig. 6f). In the control analyses, we examined how transcriptional regulation is connected to BA and AB transitions that were common

OIS and RS in IMR90 cells ($P = 2.71 \times 10^{-973}$, hypergeometric distribution test; Fig. 6c) and upon RS in IMR90 and BJ cells ($P = 2.74 \times 10^{-227}$, hypergeometric distribution test; Fig. 6d). We also observed significant overlap of AB transitions upon OIS and RS in IMR90 cells ($P = 6.01 \times 10^{-86}$, hypergeometric distribution test; Supplementary Fig. 6m) and RS in IMR90 and BJ cells ($P = 1.51 \times 10^{-35}$, hypergeometric distribution test; Supplementary Fig. 6n). These results indicate that the observed compartmental reorganizations upon RS and OIS are non-random events and, more importantly, that a significant fraction of BA transitions is conserved between the two types of senescence.

**Fig. 6 Detection of common compartmental reorganizations upon OIS and RS. a** Contact maps for chromosome 4 in IMR90 RS (top right) and growing (bottom left) cells at 200 kb resolution. Histone H3K9me3 distribution in IMR90 growing cells is shown at top. PCA scores along chromosome 4 for RS and growing cells (G) are shown at right. **b** Contact maps for chromosome 4 in BJ RS (top right) and growing (bottom left) cells at 200 kb resolution. **c** Overlap of genomic regions undergoing BA transitions upon OIS and RS in IMR90 cells. Genomic regions undergoing BA transitions that were common between the IMR90 OIS biological replicates were compared to those shared between the IMR90 RS biological replicates (hypergeometric distribution test). **d** Overlap of genomic regions undergoing BA transitions upon RS in IMR90 and BJ cells. Genomic regions undergoing BA transitions that were common between the IMR90 RS biological replicates were compared to those shared between the BJ RS biological replicates (hypergeometric distribution test). **e** GSEA of genes ($n = 811$) located at genomic regions undergoing BA transitions that were common between OIS and RS in IMR90 cells. All human genes were ranked by expression fold changes (FC), calculated as levels of gene expression in RS (left)[68] or OIS cells (right) divided by those in growing cells (blue distributions). **f** GSEA of genes ($n = 362$) located at genomic regions undergoing BA transitions upon RS that were common between IMR90 and BJ cells. **g** ChIP-seq enrichment of the indicated proteins and epigenetic marks at genomic regions in the respective compartmental categories (AA, BB, AB, and BA). Genomic regions that belonged to the same compartmental categories in IMR90 OIS and RS cells were analyzed as in Fig. 4c. G and S indicate growing and senescent (OIS and RS) cells, respectively. ChIP-seq data were derived from IMR90 growing and OIS cells. For boxplots central bar represents the median with boxes indicating the upper and lower quartiles, and whiskers extend to the data points, which are no more than 1.5× the interquartile range from the box. **h** An example showing the BA transition conserved between OIS and RS. PCA scores are plotted along a 5 Mb region in chromosome 6. **i** Same as panel **h**, but a 5 Mb region in chromosome 18 is shown. The AB transition conserved upon OIS and RS is centered.

between the RS biological replicates and observed that the common BA and AB transitions upon RS were consistently coupled to the up- and downregulation of genes, respectively, in the different cell lines (Supplementary Fig. 6i-l). These results suggest that BA and AB compartmental transitions that are conserved between OIS and RS are coupled to the up- and downregulation of genes, respectively.

Next, to understand how the compartmental reorganizations that are shared between OIS and RS are regulated at the chromatin level, we examined the binding of the genome-organizing machinery and epigenetic marks at genomic regions in the respective compartmental categories (AA, BB, AB, and BA), which were conserved between OIS and RS. CAP-H2 condensin enrichment was elevated at BA-switching regions in senescent cells compared to growing cells; and CAP-H2 enrichment was diminished in AB-switching regions in senescent cells compared to growing cells (Fig. 6g). SMC1 cohesin showed a similar binding profile, but to a lesser extent. As expected, the H3K4me3 euchromatic and H3K9me3 heterochromatic epigenetic marks were enriched at AA and BB regions, respectively, and H3K27me3 and LMNB1 (Lamin B1) were most enriched at BA-switching regions (Fig. 6g). Two examples showing individual BA and AB transitions conserved between OIS and RS are presented in Fig. 6h, i, along with corresponding ChIP-seq data showing that CAP-H2 was more enriched at the conserved BA-switching region in senescent cells compared to growing cells, but not enriched at the conserved AB-switching region (Fig. 6g–i). These results collectively suggest that condensin binding and dissociation potentially participate in BA and AB transitions, respectively, and this mechanism is likely conserved between OIS and RS.

**Condensin depletion impairs senescence markers.** Our data indicated that BA transitions that were shared between OIS and RS correlated with the enhanced enrichment of CAP-H2 condensin in senescent cells (Fig. 6g). To gain a deeper understanding of this correlation between condensin and cellular senescence, we examined how CAP-H2 condensin KD affects the senescent state; the general approach is as described in Fig. 7a. Knockdown of CAP-H2 in OIS cells significantly reduced both the number of SAHF, from ~80% to ~30% (Fig. 7b), and the percent of cells staining positive for the senescence marker SA-β-gal, from ~80% to ~40% (Fig. 7c). However, the percent of cells that incorporated BrdU remained constant across OIS and CAP-H2 KD cells (Fig. 7d). We performed IF microscopic experiments to visualize mitotic spindles and observed that OIS and CAP-H2 KD cells did not show spindle staining, indicating that CAP-H2

KD neither promotes cell-cycle re-entry nor causes mitotic defects (Supplementary Fig. 7). These data suggest that CAP-H2 condensin depletion impairs the senescent markers but is not sufficient to reverse the senescent state.

**Role of condensin in the upregulation of senescence genes.** To gain a greater understanding of how condensin controls gene expression, we evaluated the expression of senescence genes before and after condensin KD in OIS cells, using RT-qPCR and RNA-seq analyses. We found that the expression of senescence genes such as SASP and p53 target genes was consistently downregulated by CAP-H2 KD, whereas the expression of proliferation genes (i.e., E2F targets) was either up- or downregulated (Fig. 8a–c; Supplementary Fig. 8). Principal component analysis (PCA) on the RNA-seq data revealed that growing and CAP-H2 KD cells were more closely positioned along the PC2 axis compared to the OIS samples, implying that the expression profiles in growing and CAP-H2 KD cells were more correlated with each other than with those in OIS cells, in one aspect (Fig. 8d). In support of this observation, we also found that many genes ($n = 4{,}553$) upregulated upon OIS were downregulated by CAP-H2 KD (Fig. 8e). Therefore, the observed correlation between growing and CAP-H2 KD cells based on PCA scores may reflect the expression of genes that are upregulated upon OIS (downregulated in growing cells) and downregulated by CAP-H2 KD. It is noteworthy that many genes upregulated upon OIS and downregulated by CAP-H2 KD were CAP-H2 binding genes, suggesting that CAP-H2 plays a direct role in the upregulation of senescence genes (Fig. 8e).

GO analysis revealed that those genes upregulated upon OIS and downregulated by CAP-H2 KD were significantly enriched in SASP and other senescence genes (Fig. 8f). We also performed IF experiments to visualize IL1B (SASP factor) and p21 (p53 target; cell-cycle regulator) in CAP-H2 KD cells and observed that CAP-H2 KD reduced the expression of the senescence genes (Fig. 8g). In addition, western blotting results revealed that IL1B and p21 proteins were downregulated by CAP-H2 KD, while RB phosphorylation and Cyclin A (E2F target) were not affected by CAP-H2 KD (Supplementary Fig. 9). These results collectively suggest that CAP-H2 KD impairs the p53/p21 pathway but does not compromise the p16/RB pathway, which inhibits E2F activation. It is well-established that senescence-associated growth arrest is mediated by the p53/p21 and p16/RB pathways[50]. Our results indicate that the p16/RB pathway retained in CAP-H2 KD cells accounts for the observed maintenance of growth arrest in CAP-H2 KD cells, although CAP-H2 KD downregulates many

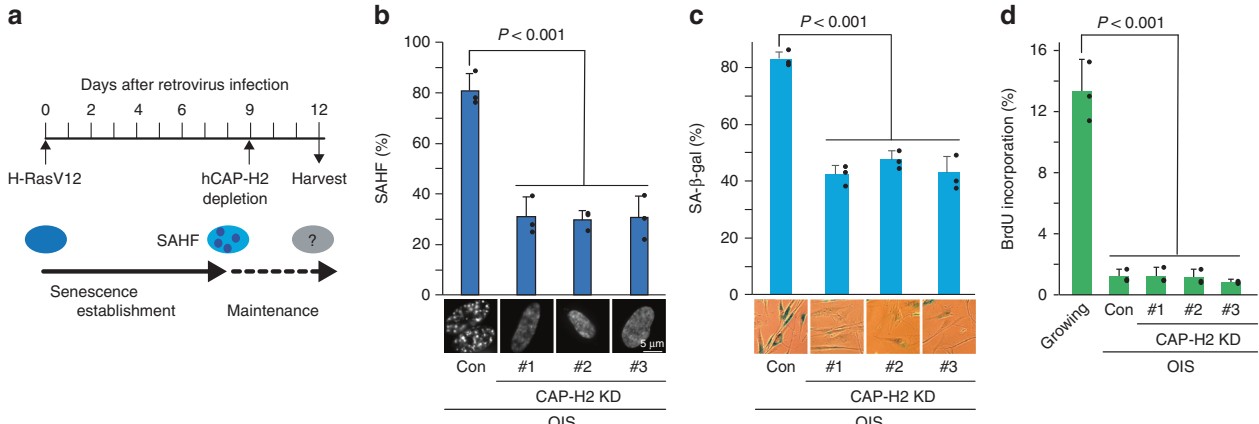

**Fig. 7 Effects of CAP-H2 condensin depletion on senescence markers. a** Schematic procedure to investigate effects of CAP-H2 condensin KD on the maintenance of OIS. IMR90 cells were infected with retrovirus encoding H-RasV12 and cultured for 9 days to establish OIS (Methods). OIS cells were then infected with lentivirus encoding one of the three shRNA constructs (#1, #2, and #3) against CAP-H2 gene (*NCAPH2*) or carrying an empty vector (control) and harvested 3 days after the lentivirus infection. **b–d** Effect of CAP-H2 KD on maintenance of OIS. Cells prepared as in panel **a** were subjected to DAPI (**b**), SA-β-gal (**c**), and BrdU (**d**) staining. *P*-values were calculated by two-sided Student's *t*-test, using biologically independent samples ($n = 3$, error bars represent the SD).

senescence genes including SASP and p53 target genes and impairs senescence markers.

To gain mechanistic insights into condensin loading to senescence genes, we focused on p53 targets, because they were the top-ranked condensin-binding genes in the GO analysis (Fig. 1e). p53 binding to its target genes was maintained after CAP-H2 KD in OIS cells, whereas CAP-H2 binding to p53 target genes was compromised by p53 KD, indicating that condensin functions downstream of p53 (Fig. 8h, i). These results suggest that condensin binding is dependent upon transcription factors including p53 and functions in the upregulation of senescence genes, presumably explaining why condensin KD impairs the senescence markers (Fig. 8j).

**Contact maps from CAP-H2 condensin KD cells**. It is possible that condensin mediates BA transitions upon OIS, which in turn enhance expression of senescence genes (Fig. 8j). To test this possibility, we investigated how CAP-H2 depletion affects 3D genome organization in OIS cells. We applied in situ Hi-C to CAP-H2 KD cells prepared as described in Fig. 7a and observed that global contact patterns in OIS and CAP-H2 KD (#1-#3) cells showed striking similarities; all contact patterns displayed the SAHF-derived contact patterns connecting heterochromatic domains (Fig. 9a, b; Supplementary Fig. 10; Supplementary Fig. 11a, b). Note that the efficiency of CAP-H2 KD (#1-#3) was confirmed (Supplementary Fig. 1a, b). We did not find a clear difference in PCA scores between the OIS and CAP-H2 KD cells (Fig. 9c; Supplementary Fig. 10; Supplementary Fig. 11c, d). However, LVS scores (long-range contacts > 2 Mb) derived from SAHF-forming genomic regions were lower in CAP-H2 KD cells compared to OIS cells, suggesting that SAHF organization is partly compromised by CAP-H2 KD (Supplementary Fig. 11e). We also observed that CAP-H2 KD slightly reduced contact probabilities between regions separated by >1 Mb (Supplementary Fig. 11f). This reduction of contact probabilities in the 'long' range (>1 Mb) by CAP-H2 KD may explain the decrease in LVS scores (Supplementary Fig. 11e) and also the decrease in the percent of cells showing SAHF upon CAP-H2 KD (Fig. 7b). These results suggest that condensin tends to facilitate >1 Mb long-range contacts in OIS cells (Supplementary Fig. 11f).

**B-to-A transitions upon OIS reversed by condensin KD**. To understand how condensin participates in compartmental reorganizations upon OIS, we examined compartmental reorganizations by CAP-H2 depletion in OIS cells. We observed that the OIS and CAP-H2 KD data showed an expansion of A compartments (~60%) compared to growing cells (~50%) (Supplementary Fig. 11g, compared to Fig. 2d). When CAP-H2 was knocked down in OIS cells, only a small percentage of the genome, 1.63% (43 Mb) and 2.29% (60 Mb), underwent BA and AB transitions, respectively (Supplementary Fig. 11g). Consistently, the average sizes and numbers of A and B compartments remained roughly constant. Those genomic regions that underwent BA or AB transitions upon CAP-H2 KD were usually 50–200 kb in size (Supplementary Fig. 11h). More importantly, AB transitions ($n =$ 419) occurred more frequently than BA transitions ($n = 289$; Supplementary Fig. 11h). These AB transitions caused by CAP-H2 KD significantly overlapped with BA transitions upon OIS ($P = 4.71 \times 10^{-1302}$; hypergeometric distribution test; Fig. 9d, e; Supplementary Fig. 11i). These results suggest that condensin is required for the maintenance of A compartments that were switched from B compartments upon OIS. BA and AB transitions that occurred at the same genomic regions upon OIS and by CAP-H2 KD were referred to as BAB transitions (Fig. 9d). We applied GSEA to genes present at genomic regions undergoing BAB transitions and found that BAB regions were significantly enriched with upregulated genes upon OIS and with down-regulated genes by CAP-H2 KD (Fig. 9f, g). These results suggest that BAB compartmental reorganizations are coupled to the transcriptional regulation, and, more importantly, that BA transitions are at least partly mediated by condensin.

We also examined how CAP-H2 KD affects TAD borders and observed that sizes and numbers of TADs in CAP-H2 KD cells were similar to those in OIS cells (Supplementary Fig. 5d). Some TAD borders were present only in OIS or CAP-H2 KD cells (Supplementary Fig. 11j), and G-specific TAD borders defined in Fig. 5a significantly overlapped with CAP-H2 KD-specific borders ($P = 3.78 \times 10^{-80}$; hypergeometric distribution test; Supplementary Fig. 11k), indicating that some TAD borders diminished upon OIS reappeared in CAP-H2 KD cells.

**B-to-A-to-B transitions occur at or near heterochromatin**. BA transitions that occurred upon OIS were not always reversed by

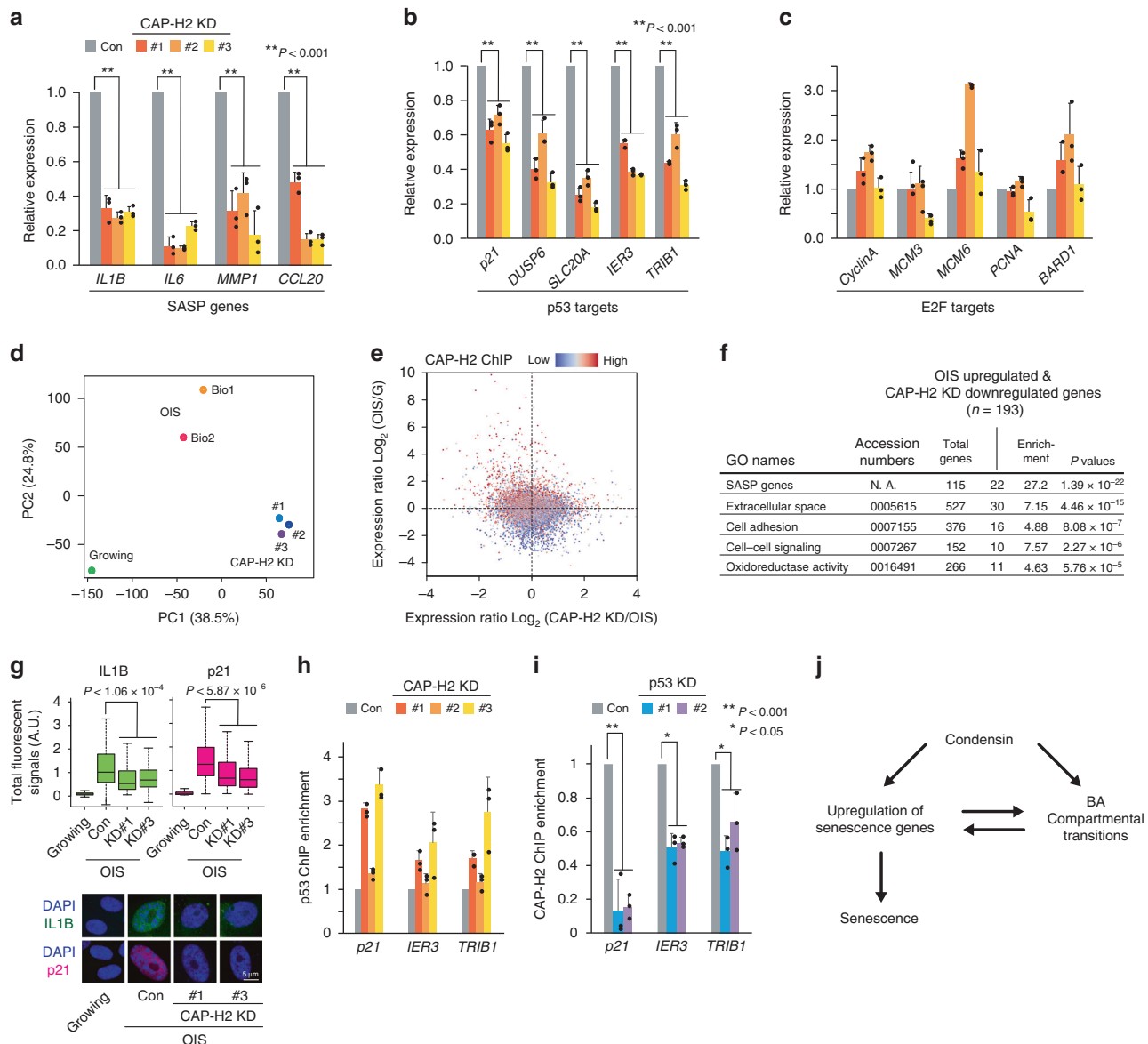

**Fig. 8 Effects of CAP-H2 condensin depletion on gene regulation in OIS cells. a–c** Effect of CAP-H2 KD on mRNA levels of the SASP (**a**), p53 target (**b**), and E2F target genes (**c**). RNA levels of the indicated genes in OIS (control) and CAP-H2 KD (#1-#3) cells were quantified by RT-qPCR. *P*-values were calculated by two-sided Student's *t*-test, using biologically independent samples ($n = 3$, error bars represent the SD). **d** PCA of RNA-seq data showing similarities and differences in global expression profiles among growing, OIS (Bio1 and Bio2) and CAP-H2 KD (#1-#3) IMR90 cells. The positioning of growing and CAP-H2 KD cells at a similar location along the PC2 axis indicates that gene expression profiles in these cell populations are correlated. PC1, PC2, principal components 1 and 2. **e** Correlation between genes that were upregulated upon OIS and downregulated by CAP-H2 KD. Expression ratios between OIS and growing cells were compared to those between CAP-H2 KD and OIS cells. Upper left quadrant indicates 4553 genes that were upregulated by OIS and downregulated by CAP-H2 KD. Dot colors reflect CAP-H2 ChIP-seq enrichment scores. **f** GO analysis of genes ($n = 193$) significantly upregulated upon OIS and downregulated by CAP-H2 KD, showing enrichment in SASP and other senescence genes (Fisher exact test). **g** Immunofluorescent (IF) visualization of IL1B (SASP factor) and p21 (p53 target; cell-cycle regulator) in OIS and CAP-H2 KD cells. Nuclear IF signals were quantified in more than 100 cells (Methods), and distributions of nuclear IF signals shown as boxplots (central bar represents the median with boxes indicating the upper and lower quartiles, and whiskers extend to the data points, which are no more than 1.5× the interquartile range from the box) were compared between CAP-H2 KD and control cells (two-sided Mann–Whitney *U* test). **h** Effect of CAP-H2 KD on p53 binding at p53 target genes in IMR90 OIS cells, as determined by ChIP-qPCR. **i** Effect of p53 KD on CAP-H2 binding at the indicated p53 target genes in OIS cells. Cells were prepared as in Fig. 7a except that CAP-H2 KD was replaced by p53 KD. *P*-values were calculated by two-sided Student's *t*-test, using biologically independent samples ($n = 3$, error bars represent the SD). **j** A model explaining how condensin may mediate gene regulation and compartmental reorganization during senescence processes. A hypothesis is that condensin participates in the upregulation of senescence genes and BA transitions, both of which are not completely independent to each other.

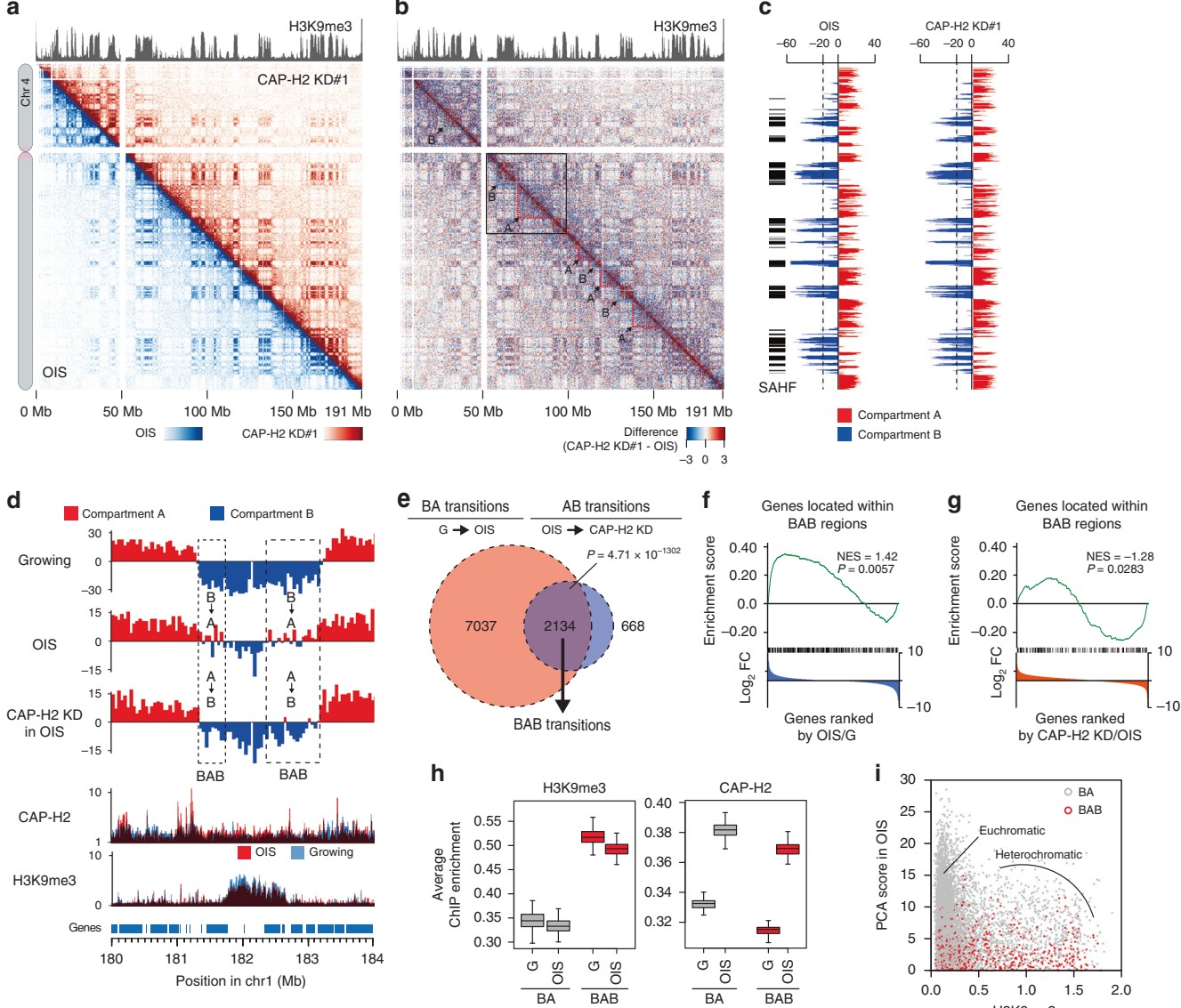

**Fig. 9 Effects of CAP-H2 depletion on A/B compartmental organizations. a** Contact maps for chromosome 4 in CAP-H2 KD (#1) (top right) and OIS (bottom left) cells at 200 kb resolution. Cells were prepared as in Fig. 7a. Histone H3K9me3 ChIP-seq data (GEO accession#, GSE38448) are shown at top. **b** Difference of contact probabilities for the data shown in panel **a**. Red and blue dots indicate that contact probabilities were higher in CAP-H2 KD and OIS cells, respectively. Red and blue dotted lines represent reduced and elevated contact scores in A and B compartments, respectively. The 50 Mb genomic region indicated by open box is enlarged in Fig. 10b. **c** PCA scores in OIS and CAP-H2 KD cells plotted along chromosome 4. SAHF were defined as genomic regions with PCA scores <−20; locations are indicated by black bars at left. **d** An example of the BAB compartmental transitions. The BAB transitions indicate regions that underwent BA transitions upon OIS and AB transitions upon CAP-H2 KD. PCA scores in the indicated cell types are plotted along a 4 Mb region in chromosome 1. CAP-H2 and H3K9me3 ChIP-seq data are shown beneath. **e** Overlap of genomic regions (40 kb bins) undergoing BA transitions upon OIS and AB transitions upon CAP-H2 KD (hypergeometric distribution test). **f** GSEA of genes (n = 395) located at genomic regions undergoing BAB transitions. Human genes were ranked by expression fold changes (FC), where expression levels of genes in OIS cells were divided by those in growing cells. **g** GSEA of genes (n = 232) located at genomic regions undergoing BAB transitions and upregulated upon OIS. Human genes were ranked as expression levels in CAP-H2 KD cells were divided by those in OIS cells. **h** Average ChIP-seq enrichment of CAP-H2 and H3K9me3 at BA and BAB regions. Genomic regions that belonged to the compartmental categories (BA and BAB) were analyzed as in Fig. 4c. For boxplots central bar represents the median with boxes indicating the upper and lower quartiles, and whiskers extend to the data points, which are no more than 1.5× the interquartile range from the box. **i** PCA scores of 40 kb bins from BA (gray) or BAB (red) genomic regions plotted against H3K9me3 ChIP-seq enrichment in OIS cells.

CAP-H2 KD, implying that the BAB transitions induced by CAP-H2 KD have some preference. To identify what might control compartmental preference, we characterized H3K9me3 enrichment at BAB transition regions induced by CAP-H2 KD in OIS cells. We observed that H3K9me3 enrichment was higher at BAB regions compared to BA-switching regions (Fig. 9h). In contrast,

CAP-H2 enrichment at both BA-switching regions and BAB regions was elevated in OIS cells (Fig. 9h). Plotting PCA scores of BA-switching regions and of BAB regions against H3K9me3 enrichment indicated that a main population of BA-switching regions were derived from euchromatic regions that showed low H3K9me3 enrichment, whereas BAB regions were derived from

genomic regions having a spectrum of H3K9me3 enrichment scores, likely representing heterochromatin domains (Fig. 9i). These results suggest that BA transitions upon OIS frequently, but not always, occur at euchromatic or weak heterochromatic regions, whereas BA transitions that occur at or near heterochromatic regions tend to be reversed when CAP-H2 condensin is depleted, resulting in BAB transitions. This is likely because heterochromatin present at or near BA-switching regions can promote an AB transition when the region is not protected by condensin, suggesting that condensin is involved in maintaining euchromatic A compartments in senescent cells (Examples in Fig. 9d).

**Roles of B-to-A transitions upon OIS in gene regulation**. To understand what role BA transitions might play in gene regulation, we compared expression levels of significantly upregulated genes present at BA-switching regions with expression levels of all significantly upregulated genes, derived mainly from AA regions. We found that expression ratios between OIS and growing cells were higher for genes present at BA-switching regions compared to all significantly upregulated genes (control), indicating that genes at BA regions were upregulated to a greater degree upon OIS than were other genes (Fig. 10a). CAP-H2 KD reduced the expression of genes at BA regions more than it reduced the expression of all upregulated genes (Fig. 10a). These results

suggest that a function of BA transitions upon OIS is to enhance the activation levels of genes within these regions compared to genes at other genomic regions, and that condensin is involved in maintaining gene expression at the BA-switching regions more than at other regions.

**Potential roles of condensin at euchromatic domains**. Given that condensin binding is involved in the upregulation of senescence genes (Fig. 8e) and that significantly upregulated genes ($n = 842$) are mainly derived from AA regions ($n = 631$; Fig. 10a), it was possible that condensin mediates genomic contacts and the upregulation of genes in A compartments. To test this possibility, we compared the contact maps from OIS and CAP-H2 KD cells and observed that contact scores in A and B compartments were significantly reduced and enhanced, respectively, in CAP-H2 KD cells compared to OIS cells ($P < 0.001$, two-sided Mann–Whitney $U$ test; Fig. 10b, c). The bottom 10% of A compartments (i.e., compartments with the lowest relative contact scores between CAP-H2 KD and OIS cells) had significantly enhanced CAP-H2 enrichment upon OIS ($P = 3.21 \times 10^{-8}$, two-sided Mann–Whitney $U$ test; Fig. 10d). Moreover, GSEA revealed that genes present in the bottom 10% of A compartments tended to be downregulated by CAP-H2 KD (Fig. 10e). These results suggest that condensin binding to genes in A compartments

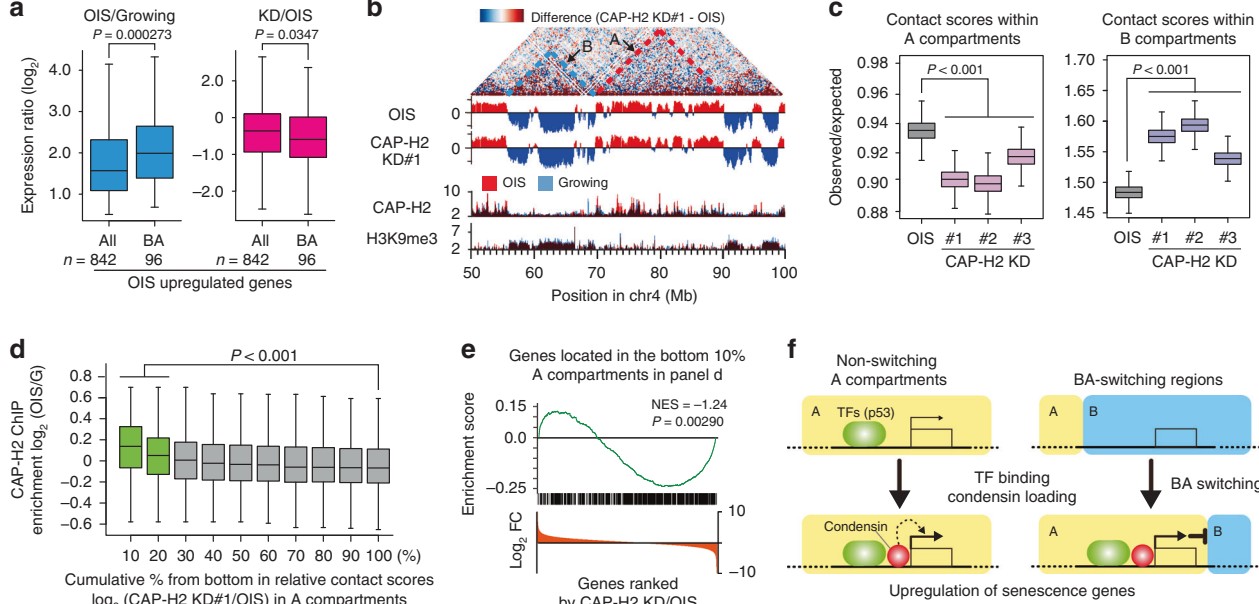

**Fig. 10 Condensin reinforces euchromatic A compartments. a** Left: Expression ratios between OIS and growing cells for all genes that were significantly upregulated genes ($n = 842$) upon OIS, and for significantly upregulated genes present at BA-switching regions ($n = 96$) shown as boxplots (central bar represents the median with boxes indicating the upper and lower quartiles, and whiskers extend to the data points, which are no more than 1.5× the interquartile range from the box). $P$-values were calculated by a two-sided Mann–Whitney $U$ test. Right: The same sets of genes were used to calculate expression ratios between CAP-H2 KD and OIS control cells. **b** An example showing that contact scores in the A compartment were slightly reduced in CAP-H2 KD cells compared to OIS cells (red dotted line). Blue dotted line indicates that contact scores in the B compartment were relatively elevated by CAP-H2 KD. Difference map between OIS and CAP-H2 KD cells shows a 50 Mb genomic region in chromosome 4. PCA scores and ChIP-seq data (CAP-H2 and H3K9me3) are shown beneath. **c** Effect of CAP-H2 KD on contact scores within A and B compartments. Distance-normalized contact scores at 200 kb resolution were calculated by observed contact scores divided by expected scores estimated from all combinations separated by same distances. Boxplots are shown as in panel **a**, and $P$-values were calculated by a two-sided Mann–Whitney $U$ test. **d** CAP-H2 relative enrichment (in OIS over growing cells) in A compartments ranked by relative contact scores between CAP-H2 KD and OIS cells. Distance-normalized contact scores were used for this analysis. Boxplots are shown as in panel **a**, and $P$-values were calculated by a two-sided Mann–Whitney $U$ test. **e** GSEA of genes ($n = 710$) located in the bottom 10% of A compartments with the lowest relative contact scores between CAP-H2 KD and OIS cells (data shown in panel **d**) and with elevated CAP-H2 enrichment in OIS cells compared to growing cells. Human genes were ranked by expression fold changes (FC), where expression levels of genes in CAP-H2 KD cells were divided by those in OIS cells. **f** A model to explain how condensin may reinforce non-switching A compartments (left) and promote/maintain BA compartmental transitions during senescent processes (right; see Discussion for details).

positively contributes to not only genomic contacts but also the upregulation of genes (Fig. 10f).

To better understand condensin-mediated compartmental reorganizations predicted from the in situ Hi-C genomic data, we sought to obtain cell biological insights into how condensin behaves within the human nucleus. We examined the subnuclear localization of CAP-H2 proteins in OIS cells and found that CAP-H2 and RNA Pol II were localized at regions surrounding SAHF, suggesting that condensin II is present in the euchromatic nuclear compartment (Supplementary Fig. 12a,b). Additionally, the CAP-H2 localization pattern represented a mixture of speckled and potential filamentous distributions, with the filamentous distributions being reminiscent of subnuclear architecture (Supplementary Fig. 12c,d). Nascent transcripts and CAP-H2 were preferentially detected outside SAHF (Supplementary Fig. 12e). When we performed fractionation analysis we found that CAP-H2 proteins existed in the chromatin-bound fraction (F3), where histone H3 was observed (Supplementary Fig. 12f; Supplementary Methods). CAP-H2 proteins were also detected in the insoluble nuclear fraction (F4), where lamin B1 accumulated (Supplementary Fig. 12f). These results may suggest that condensin II not only associates with euchromatic DNA but also is a potential constituent of the nuclear architecture (See Discussion).

## Discussion

The SMC complex cohesin and CTCF have been shown to mediate the formation of TADs[7,13,51,52]. Disruption of TADs does not impact the global organization of A and B compartments, suggesting that the organization of TADs and compartments are largely independent[53]. Here, our study has suggested that another SMC complex, condensin, is at least partly involved in the compartmental reorganization that occurs during cellular senescence.

We previously showed that overexpression of condensin II, but not condensin I, induces cellular senescence and SAHF formation, and that the establishment of OIS is inhibited by condensin II depletion[43]. Given that senescent processes are accompanied by global alterations in the 3D genome architecture[19,20], the first goal of the study described here was to understand the global reorganization of the human genome upon senescence. Unexpectedly, our 3D genomic data show that roughly 14% of the human genome (430 Mb out of 3100 Mb) undergoes a BA transition upon OIS, revealing that euchromatic A compartments expand in non-proliferating OIS cells compared to proliferating cells. Consistent with this observation, nascent RNA levels are elevated in OIS cells compared to growing cells (Supplementary Fig. 2k). Genome-wide occupancies of A compartments increase upon OIS and RS, suggesting that the enlargement of euchromatic A compartments may be a common senescent feature conserved between OIS and RS. Although SAHF formation upon OIS is not a conserved senescent phenomenon nor has it been observed in endogenous tissues[20,48], BA-switching regions upon OIS significantly overlap with those upon RS, demonstrating that BA transitions are at least partly conserved between OIS and RS.

In efforts to identify a functional link between senescence-dependent compartmental reorganizations and condensin, we find that the condensin II-specific subunit, CAP-H2, is enriched in euchromatic A compartments. Condensin enrichment is elevated at BA-switching regions, and condensin KD often converts A compartments at BA-switching regions to B compartments, suggesting that condensin not only mediates BA transitions but also is required for the maintenance of A compartments in

BA-switching regions (Fig. 10f). It has been appreciated that condensin is recruited to its locations by various binding factors[54,55]. Here, we find that condensin functions downstream of p53, as p53 KD impairs condensin localization at its target genes, but not vice versa. Therefore, it is likely that senescence genes located in B compartments are initially bound by transcription factors including p53, which in turn recruit condensin, leading to BA transitions upon senescence (Fig. 10f).

In addition to promoting and maintaining BA transitions, condensin facilitates genomic contacts in A compartments (Fig. 10c, d). This result suggests that an important role of condensin is to reinforce A compartments (Fig. 10f). It was previously reported that cohesin and condensin components are present in a human nuclear insoluble fraction[56]. Our studies confirm this and show that condensin partly resides in the nuclear insoluble fraction in OIS cells (Supplementary Fig. 12)[43]. Therefore, our current hypothesis is that the condensin-localized nuclear architecture helps separate A compartments from B compartments via fixing active genes on the architecture. Because the condensin-localized structure was only examined in OIS cells, it remains unclear if it is formed in RS cells. It is also possible that condensin plays an active role in the upregulation of senescence genes and housekeeping genes, promoting the stabilization of A compartments indirectly.

Our results show that the full expression of p53 target genes such as *p21* and SASP genes such as *IL1B* is condensin-dependent at both transcription and protein levels. Therefore, condensin positively contributes to senescent processes. However, condensin KD in OIS cells cannot reverse the senescent state, even though condensin KD downregulates SASP and p53 target genes. In this regard, it is shown that cellular senescence is mediated by the p53/p21 and p16/RB pathways[50], and we find that condensin KD does not affect the p16/RB pathway, which inhibits E2F activation. Our results support a model whereby condensin does not participate in the p16/RB pathway, which accounts for the observed maintenance of growth arrest in condensin KD cells. This study shows that condensin is necessary for the full activation of SASP and the p53/p21 pathway and that condensin KD impairs senescence, whereas condensin KD does not affect the p16/RB pathway, which maintains the growth arrest observed in established senescent cells with condensin KD.

Our results suggest that p53 functions upstream of condensin. p53 probably mediates basal expression of its target genes, and condensin promotes the expression of p53 targets by potentially mediating specific 3D genomic contacts. In this regard, it has been shown that condensin II binds to enhancers and promoters and mediates gene activation[32,33]. Our current hypothesis is that condensin mediates the full activation of senescence genes via enhancer-promoter contacts, via local chromatin regulation involving its direct binding to gene promoters, and/or via BA transitions (Fig. 8j).

All together, this study suggests that the roles of condensin are to reinforce euchromatic A compartments and to promote/maintain BA transitions upon senescence, both of which are connected to the upregulation of senescence genes (Fig. 10f). Therefore, condensin participates in cellular senescence through compartmental reorganizations coupled to gene regulation.

## Methods

**Preparation of OIS cells.** Normal diploid IMR90 human fibroblasts were cultured in DMEM medium supplemented with 10% FBS, 100 U/ml penicillin, 100 μg/ml streptomycin, 0.15% sodium bicarbonate, 2 mM L-glutamine, 1 mM sodium pyruvate, and 1 x MEM non-essential amino acids (Thermo Fisher Scientific, 11120052). Growing cells in population doublings (PD) between 26 and 36 were used for experiments. pBABE-puro plasmids carrying H-*ras*V12 were used for retrovirus packaging, and virus infection was performed as described previously[57].

H-*ras*V12-expressing cells were cultured in medium containing 1 μg/ml puromycin for 7–10 days to obtain OIS cells.

**Preparation of RS cells**. IMR90 cells were cultured in DMEM medium supplemented with 10% FBS, 100 U/ml penicillin, 100 μg/ml streptomycin, 0.15% sodium bicarbonate, 2 mM L-glutamine, 1 mM sodium pyruvate, and 1 x MEM non-essential amino acids. PD 32 growing and PD 84 senescent IMR90 cells were kindly gifted from the Peter Adams laboratory (Sanford Burnham Prebys Medical Discovery Institute). BJ human fibroblasts were obtained from ATCC and cultured in DMEM medium supplemented with 10% FBS, 100 U/ml penicillin, and 100 μg/ml streptomycin. PD 35 growing and PD 88 senescent BJ cells were employed for experiments.

**Detection of senescence markers**. SA-β-gal staining was performed as previously described[58]. Cells grown on coverslip were washed twice with 1 ml PBS and fixed at room temperature for 5 min in 1 ml PBS containing 2% formaldehyde and 0.2% glutaraldehyde. After washing cells twice with 1 ml PBS, cells were incubated with 500 μl staining solution [40 mM $Na_2PO_4$ pH 6.0, 150 mM NaCl, 2 mM $MgCl_2$, 5 mM $K_3Fe(CN)_6$, 5 mM $K_4Fe(CN)_6$, 1 mg/ml X-gal] at 37 °C overnight in dark. BrdU staining was performed as previously described[43]. Cells grown on coverslip were labeled with 100 μg/ml BrdU (Sigma–Aldrich, B5002) for 30 min. Cells were subsequently fixed with 4% paraformaldehyde (pFA) for 15 min, washed three times with 1 ml PBS, and treated with 0.5% Triton-X100 in PBS at 4 °C for 5 min. Cells were washed three times with 1 ml PBST (PBS containing 0.05% Tween 20) and incubated with 2 N HCl at room temperature for 10 min. Cells were again washed three times with 1 ml PBST and soaked in 1 ml PBS containing 1% BSA at room temperature for 10 min. Cells were then incubated with 1:10-diluted FITC-labeled mouse monoclonal anti-BrdU (BD Biosciences, 556028) for 30 min. After washing cells three times with 1 ml PBST, nuclei were stained with 1 μg/ml DAPI solution for 5 min. In order to detect SAHF, cells were fixed with 1% pFA, and nuclei were stained with 1 μg/ml DAPI solution for 5 min.

**Knockdown of CAP-H2 and p53**. pTRIPZ plasmids containing *NCAPH2* shRNAs (#1, V3THS_326285; #2, V3THS_326286; #3, V3THS_326290, Dharmacon) were used for CAP-H2 knockdown. pLKO1 plasmids containing *p53* shRNAs (#1, TRCN0000003753; #2, TRCN0000003755, Sigma–Aldrich) were used for p53 knockdown. Lentivirus was prepared using ViraPower kit (Thermo Fisher Scientific) according to the manufacturer's protocol. Doxycycline (1 μg/ml) was added to culture medium every 48 h after lentiviral transfection for shRNA expression.

**Exogenous expression of CAP-H2-FLAG**. pCSII-EF-puro plasmid expressing FLAG-tagged CAP-H2 protein, which was under the control of the EF-1α promoter, was constructed and used for lentivirus production. Cells were used for experiments 72 h after transfection.

**In situ Hi-C**. In situ Hi-C was performed as previously described[9] with slight modifications. Cells at 80% confluence in 10 cm dish were fixed in 1% pFA at room temperature for 10 min and subjected to the in situ Hi-C procedure. After proximal ligation, ligation products were enriched by centrifugation at 10,000 rpm at 4 °C for 10 min. The pellet was suspended with proteinase K solution consisting of 10 mM Tris-HCl, pH 8.0, 50 mM EDTA, and 2 mg/ml proteinase K (Thermo Fisher Scientific) and incubated at 37 °C for 1 h, followed by overnight incubation at 68 °C with 0.5 M NaCl. DNA was purified by phenol/chloroform extraction and sonicated at 4 °C using Bioruptor (Diagenode) for 15 min by repeating the cycle of 30-sec ON and 1-min OFF. Biotin-labeled DNA was purified using Dynabeads (MyOne Streptavidin T1, Thermo Fisher Scientific). Sequencing adapters were ligated using NEBNext Ultra Ligation module (New England Biolabs, Inc) on Dynabeads, and DNA was PCR-amplified for 8–14 cycles using NEBNext Q5 Hot Start HiFi PCR master Mix and NEBNext multiplex oligos (New England Biolabs, Inc). PCR products were sequenced on Illumina NextSeq 500 platform to obtain 76-bp paired-end reads.

**Processing in situ Hi-C data**. In situ Hi-C data were processed as described[31], and the highest resolutions were defined as previously described (Supplementary Table 1)[59]. Briefly, 76-bp paired reads were separately aligned to the human genome (hg19) using Bowtie2 (version 2.2.9) with iterative alignment strategy. Redundant paired reads derived from a PCR bias, reads aligned to repetitive sequences, and reads with low mapping quality (MapQ < 30) were removed. Reads potentially derived from self-ligation and undigested products were also discarded. Hi-C biases in contact maps were corrected using the ICE method[60]. The ICE normalization was repeated 30 times. Read numbers remaining after the respective filtering processes were summarized in Supplementary Table 1.

**Calculation of PCA scores**. PCA scores were calculated using ICE normalized contact matrixes at 40 kb resolution, as previously described[5]. The signs of PCA score were determined by gene density. A and B compartments were defined by more than or equal to five consecutive bins (40 kb) showing positive and negative PCA scores, respectively. PCA scores <−20 were defined as SAHF regions.

**Detection of TAD borders**. TAD borders were defined using border strength scores as previously described[26,30]. Border strength scores were calculated based on ICE normalized contact matrixes at 40 kb resolution. Border strength scores were defined as follows:

$$R = \left[ \begin{array}{c} (\textit{Sum of contact scores within upstream 13 bins}) + \\ (\textit{Sum of contact scores within downstream 13 bins}) \end{array} \right] \div \\ (\textit{Sum of contact scores between upstream and downstream 13 bins}), \quad (1)$$

Average $R$ value across the genome was subtracted from $R$ values at respective 40 kb bins, and these values were divided by the standard deviation.

**Chromatin immunoprecipitation (ChIP)**. ChIP was performed as previously described with modifications[61]. Mouse monoclonal anti-FLAG (clone M2, Sigma–Aldrich), rabbit polyclonal anti-CAP-H2 (Bethyl Laboratories, A302-275A, A302-276A), SMC2 (Bethyl Laboratories, A300-058A), rabbit polyclonal anti-SMC1 (Bethyl Laboratories, A300-055A), rabbit polyclonal anti-CTCF (Millipore, 07-729), mouse monoclonal anti-Pol II (Covance, 8WG16), and mouse monoclonal anti-p53 (Thermo Fisher Scientific, DO-7) were used. Around $1 \times 10^7$ cells were fixed with 1.5 mM ethylene glycol-bis(succinic acid N-hydroxysuccinimide ester) in PBS at room temperature for 30 min. Cells were further fixed by 1% pFA at room temperature for 10 min and subsequently treated with 125 mM glycine for 5 min. Cells were washed with 10 ml cold PBS, harvested with 500 μl ChIP lysis buffer 1 [50 mM HEPES-KOH pH 7.5, 140 mM NaCl, 1 mM EDTA, 1% Triton X-100, 0.1% sodium deoxycholate, 0.1 mM PMSF, EDTA-free protease inhibitor cocktail (Roche)], and incubated at 4 °C for 10 min. The cell suspension was centrifuged at 3000 rpm at 4 °C for 3 min. The pellet was resuspended in 500 μl ChIP lysis buffer 2 (10 mM Tris-HCl pH 8.0, 200 mM NaCl, 1 mM EDTA, 0.5 mM EGTA, 0.1 mM PMSF, EDTA-free protease inhibitor cocktail) and incubated at room temperature for 10 min. The cell suspension was centrifuged at 3000 rpm at 4 °C for 5 min. The pellet was again resuspended in 400 μl ChIP lysis buffer 3 (10 mM Tris-HCl pH 8.0, 100 mM NaCl, 1 mM EDTA, 0.5 mM EGTA, 0.1% sodium deoxycholate, 0.5% N-lauroylsarcosine sodium, 0.1 mM PMSF, EDTA-free protease inhibitor cocktail). DNA in the cell lysate was sonicated using Bioruptor (Diagenode) by repeating the cycle of 30-sec ON and 1-min OFF for three times at a low output level. After adding 40 μl of 10% Triton X-100, final 5 mM $MgCl_2$, and subsequently final 0.25 U Benzonase (Sigma–Aldrich) to the sonicated lysate, chromatin was digested at 4 °C for 30 min, and final 10 mM EDTA was added to stop the Benzonase reaction. The Benzonase-treated sample was centrifuged at 14,000 rpm at 4 °C for 15 min. The supernatant was transferred to a new tube, and 10% of this supernatant was kept as "Input" DNA. Total 50 μl ChIP lysis buffer 1 containing 5 μl antibody and 15 μl Dynabeads protein G (Thermo Fisher Scientific) was added to the supernatant, and the bead suspension was incubated at 4 °C overnight with gentle rotation. The beads were washed five times by rotating at 4 °C for 15 min: Twice with ChIP lysis buffer 1, once with ChIP lysis buffer 1 containing 0.65 M NaCl, once with Washing buffer (10 mM Tris-HCl pH 8.0, 250 mM LiCl, 0.5% NP-40, 0.5% sodium deoxycholate, 1 mM EDTA), and once with TE (10 mM Tris-HCl pH 8.0, 1 mM EDTA). The protein-DNA mixture was eluted by incubating the beads with TES (10 mM Tris-HCl pH 8.0, 1 mM EDTA, 1% SDS) at 65 °C for 30 min and subjected to reverse cross-linking overnight at 65 °C. After treating the eluate with 1 mg/ml Proteinase K at 37 °C for 2 h, DNA was purified using QIAquick PCR purification kit (Qiagen) and eluted with EB (10 mM Tris-HCl pH 7.5). ChIP DNA was analyzed by the 7900HT fast real-time PCR system (Thermo Fisher Scientific). Primer sequences are shown in Supplementary Table 2.

**ChIP-seq**. ChIP-seq was carried out as described previously[62]. Approximately 1 ng ChIP DNA was subjected to library construction using NEBNext Ultra DNA library prep kit (New England Biolabs). Adaptor-ligated DNA was PCR-amplified for 12 cycles using NEBNext Q5 Hot Start HiFi PCR master Mix and NEBNext multiplex oligos (New England Biolabs). PCR products were sequenced on Illumina NextSeq 500 platform to obtain 76-bp single-end reads. Reads were aligned to the human genome (hg19) using Bowtie2 (version 2.2.9). Redundant reads were removed by Picard (version 2.7.1). Reads with low mapping quality (MapQ < 10) were also removed. ChIP-seq peaks were defined by HOMER software (version 4.8.3) using the option "-center -style factor -F 1 -P 0.0001 -fdr 0.05" for RNA Pol II and TP53, while the alternative option "-center -style histone -F 2 -P 0.0001 -fdr 0.05" was used for other proteins. Input DNA described above in the ChIP procedure was also subjected to the same bioinformatic analysis and employed for peak calling; Genomic sections carrying relative enrichment scores between ChIP and Input DNA samples above 2-fold, P < 0.0001 and FDR < 0.05 were defined as peaks.

**RT-qPCR**. RT-qPCR was performed as previously described[43]. RNA was extracted from IMR90 cells using RNeasy Mini kit (Qiagen) and subjected to reverse transcription using High-Capacity cDNA Reverse Transcription kit (Thermo Fisher Scientific). cDNA was used as a template in subsequent PCR with gene specific primers (Power SYBR Green master mix, Thermo Fisher Scientific). Primer sequences are shown in Supplementary Table 3. *B2M* gene encoding β2-microglobulin was used as an internal control to normalize mRNA expression

levels of other genes. PCR amplification of cDNA with gene specific primers was monitored by the 7900HT fast real-time PCR system (Thermo Fisher Scientific).

**RNA-seq.** RNA-seq was carried out as previously described with modifications[31]. Poly (A) RNA was purified using NEBNext Poly (A) mRNA Magnetic Isolation Module (New England Biolabs), and sequence libraries were constructed using NEBNext mRNA Library prep Master mix for Illumina (New England Biolabs). Adaptor-ligated DNA fragments were PCR-amplified for 12 cycles using NEBNext Q5 Hot Start HiFi PCR master Mix and NEBNext multiplex oligos (New England Biolabs). RNA-seq libraries were sequenced on Illumina NextSeq 500 platform to obtain 76-bp single-end reads.

Sequenced reads were aligned to the human genome (hg19) using the STAR program (v2.5.2)[63]. Reads assigned to exons were estimated by the RSEM program (v1.2.31)[64], and genes carrying less than 10 reads were eliminated, resulting in total 19,591 genes. Read numbers in different samples were normalized using the DEseq2 program ver3.7[65]. The DEseq2 program was used to identify differently expressed genes. CAP-H2 KD samples (KD#1 and KD#3) were treated as biological replicas. FDR < 0.05 was used as a threshold to identify significantly up- or downregulated genes.

**Gene ontology (GO) analysis.** Gene ontology (GO) analysis for CAP-H2 binding genes was done using Qiagen's Ingenuity Pathway Analysis software (IPA, QIA-GEN, www.qiagen.com/ingenuity) with "Regulator Effects" option. The GO slim groups were downloaded from http://geneontology.org/page/go-subset-guide. An additional group of SASP genes was defined previously[61]. Lists of genes were tested for the enrichment in the GO slim and SASP gene groups using Fisher exact test, and nominal P-values were corrected for multiple testing using Benjamini-Hochberg method.

**Gene set enrichment analysis (GSEA).** GSEA was carried out as previously described[46]. GSEA was performed against genes pre-ranked by relative expression levels between two selected cell types. Expression profiles in IMR90 growing, OIS, CAP-H2 KD (#1–#3) cells were derived from our RNA-seq data, while expression profiles in IMR90 RS (GSE53356) and BJ RS cells (GSE63577) were derived from the published RNA-seq data[66,67].

**Immunofluorescence (IF) microscopy.** Immunofluorescence (IF) was performed as previously described[43]. Cells grown on coverslip were fixed with 1% pFA in PBS containing 200 mM sucrose for 10 min. Cells were washed three times with 1 ml PBS and treated with 0.5% Triton-X100 in PBS at 4 °C for 5 min. Cells were again washed three times with 1 ml PBST (PBS containing 0.05% Tween 20). Cells were soaked in PBS containing 1% BSA at room temperature for 10 min and incubated for 30 min with 1:1000-diluted primary antibodies, such as mouse monoclonal anti-IL-1β (Santa Cruz Biotechnologies, E7-2-hIL1β), mouse monoclonal anti-p21 (Santa Cruz Biotechnologies, sc-817), mouse monoclonal anti-FLAG M2 (Sigma–Aldrich), rabbit polyclonal anti-Pol II (Bethyl Laboratories, A300-653A), rabbit polyclonal anti-Myc (Abcam, ab9106), and mouse monoclonal anti-Pol II (Covance, 8WG16). Cells were washed three times with 1 ml PBST and incubated with 1:1000-diluted secondary antibodies, Alexa Flour 488-conjugated anti-rabbit IgG (Thermo Fisher Scientific) and/or Cy3-conjugated anti-mouse IgG (Jackson ImmunoResearch) in PBS for 15 min. After washing cells twice with PBST, the coverslip was mounted onto slide glass using ProLong Gold antifade mountant (Thermo Fisher Scientific). Images were captured by a Zeiss Axioimager Z1 fluorescence microscope with an oil immersion objective lens (Plan Apochromat, 63 ×, NA 1.4, Zeiss).

**Quantification of IF images (for Fig. 8g).** To quantify fluorescent signals, low magnified images were captured by Nikon 80i Upright Microscope with ×20 objective lens. Nuclear area was defined by DAPI signals, and total fluorescent signals were quantified for each nucleus using NIS-Elements Microscope Imaging Software (Nikon). Fluorescent signals derived from five randomly selected circular areas outside cells were quantified for each image and used as background. Total nuclear IF signals were calculated as follows.

$$\begin{aligned} \text{Total IF signals per nucleus} = \text{Total nuclear IF signals} \\ -(\text{Nuclear area} \times \text{Average background signal density}), \end{aligned} \quad (2)$$

where nuclear area was estimated by the number of pixels with DAPI staining. Average background signal density was calculated by background signals divided by selected area (pixels). Distributions of total IF signals per nucleus were plotted as boxplots for respective samples and subjected to two-sided Mann–Whitney U test.

**Fluorescence in situ hybridization (FISH) assay.** The bacterial artificial chromosome (BAC) clones, RP11-21I19 (Probe P1: Fig. 2h; Supplementary Fig. 4h,i), RP11-70M19 (P2), RP11-80I4 (P3) and RP11-79G9 (P4), were obtained from the Children's Hospital Oakland Research Institute. To generate FISH probes, BAC DNA was sonicated using Bioruptor (Diagenode) by repeating the cycle of 30-sec ON and 1-min OFF for nine times, and 100 bp–800 bp DNA fragments were

purified using Amicon Ultra 10 K centrifugal filter (Millipore). FISH probes specific to BAC DNA fragments were prepared using Cy3-dCTP (GE Healthcare) and random primer DNA labeling kit (Takara) as described previously[36]. FISH probes (100 ng) were mixed with 20 µg Cot1 DNA (Thermo Fisher Scientific) and subjected to ethanol precipitation. The pellet was suspended in Hybridization buffer 1 (50% formamide, 2X SSC, 10% dextran sulfate), denatured at 73 °C for 5 min, and immediately chilled on ice.

FISH assay was performed as described previously[61] with modifications. Cells grown onto coverslip were washed with 1 ml PBS and soaked in 1 ml of 75 mM KCl solution for 20 min. After adding 167 µl methanol: acetic acid (3:1) fresh fixative to the KCl solution, cells were incubated for 10 min and subsequently with 1 ml fresh fixative solution at -20 °C overnight. Cells were rinsed twice with fresh fixative and steam-dried for 1 minute. After treatment with 30 µg/ml RNase A in PBS at 37 °C for 30 min, cells were washed three times with PBST. Cells were treated with 1 ml of 0.01 M HCl solution containing 0.1 mg/ml pepsin at 37 °C for 3 min and washed twice with PBST. The sample was again fixed at room temperature for 10 min with 1% pFA in PBS containing 50 mM $MgCl_2$ and washed three times with 1 ml PBST. Cells were consecutively dehydrated for 2 min by soaking coverslip in 1 ml 70% ethanol, 80% ethanol, and 100% ethanol and air-dried. Cells were incubated with 100 µl Hybridization buffer 2 (70% formamide, 2X SSC) at 73 °C for 3 min, and subjected to the consecutive dehydration treatment for 2 min with 70% ethanol, 80% ethanol and 100% ethanol and air-dried. For hybridization, the coverslip was covered by 100 µl denatured FISH probes and incubated at 37 °C overnight. Cells were washed once with 1 ml Wash buffer (50% formamide, 2 × SSC) at 43 °C for 10 min and three times with 1 ml PBST at room temperature for 5 min. DNA was stained by 1 µg/ml DAPI in PBS at room temperature for 10 min. After washing cells twice with PBST, the coverslip was mounted onto slide glass using ProLong Gold antifade mountant (Thermo Fisher Scientific). FISH images were captured by a Zeiss Axioimager Z1 fluorescence microscope with an oil immersion objective lens (Plan Apochromat, 100×, NA 1.4, Zeiss). The images were acquired at 0.4 µm intervals in the z-axis controlled by Axiovision 4.6.3 software (Zeiss). More than 100 cells were analyzed for every FISH experiment.

**Reporting summary.** Further information on research design is available in the Nature Research Reporting Summary linked to this article.

## Data availability

All sequencing data has been deposited into the Gene Expression Omnibus (GEO) under accession code GSE118494. The data that support the findings of this study are available from the corresponding author upon reasonable request.

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

## Acknowledgements

We would like to thank the Wistar Institute Genomics and Bioinformatics Facilities for high-throughput sequencing and genomic data analyses; the Wistar Imaging Facility (James E. Hayden) and the CDB Microscopy Core (Dr. Andrea Stout) at the Perelman School of Medicine at the University of Pennsylvania for microscopic analysis; and the RIKEN BioResource Research Center, Dr. Yoichi Sekita (Kitasato University) and Dr. Yusuke Shiromoto (Wistar Institute) for pCSII-EF-puro plasmid. We also thank Dr. Rachel Locke for critically reading the manuscript, and Sylvie Shaffer for editorial assistance. This work was supported by the National Institutes of Health/National Institute of General Medical Sciences (R01GM124195 to K.N.), National Institutes of

Health/National Institute on Aging (P01AG031862 to K.N. and R.Z.) and National Institutes of Health/National Cancer Institute (R01CA160331 to R.Z. and K.N.). Support for Shared Resources utilized in this study was provided by Cancer Center Support Grant (CCSG) P30CA010815 to the Wistar Institute.

## Author contributions

O.I. performed the in situ Hi-C experiments, RT-PCR, RNA-seq, ChIP, ChIP-seq, and GSEA analyses. H.T. performed the bioinformatics and statistical analyses. K.K. performed the microscopic experiments. A.K. performed the bioinformatics analyses. T.N. prepared retrovirus encoding H-RasV12. S.T. performed the biochemical experiments. S.M. prepared senescent BJ cells. L.C.S. and R.Z. helped supervise the project. K.N. conceived and designed the study and was in charge of overall direction. All authors contributed to the writing of the manuscript.

## Competing interests

The authors declare no competing interests.
