## [Peer Review File · Nature Communications]

Reviewers' comments:

Reviewer #1 (Remarks to the Author):

In this manuscript, Iwasaki and colleagues investigate the role of condensin II complexes in interphase cells that are either proliferating or have undergone senescence. There have been sporadic reports (only from mouse ESCs for mammalian systems) that SMC2 may bind and regulate promoters and enhancers, but also contradictory data from non-stem cell models (fibroblasts). The authors have also themselves shown that yeast condensin II contributes to genome organization, outside of its established role in the cell cycle. Thus, such an interrogation is interesting and could allow some resolution in the apparent SMC2 controversy in human cells, on top of the proposed role in oncogene-induced senescence.

However, the authors themselves have published on an accentuated role for SMCs and their gene associations in mitosis (Iwasaki et al, NAR, 2016) and recent published work points to the vast majority of SMC-depletion effects stemming from chromosome mis-segregation effects, rather than from direct gene regulation events -- crucially, preventing karyotype abnormalities in daughter cells restores a normal transcriptome despite condensin inactivation (Hocquet et al, eLife, 2018). All this data is of course in yeast, so it might of course be that in higher metazoans condensin II complexes have indeed acquired additional roles during interphase. Still, in *Drosophila*, the Cap-H2 subunit works with Mrg factors to exert some gene expression control -- and according to my understanding it has been formally shown that this is a complex involving the full condensin II complex. And in *C. elegans*, work from the Dekker and Meyer labs, has shown that condensin II is essentially only important for X-chromosome functions.

Interestingly, Haeuseler et al. (Genes DEv., 2008) have shown that condensin interacts with polIII-specific factors to allow their functional nucleolar clustering in yeast cells, and depletion of essentially any subunit leads to declustering and to a loss of their insulatory function for polIII genes. Then, Sutani et al. (Nat Commun., 2015) showed how condensin II binding to active promoters is a means to prime the genome for mitosis, rather than to regulate gene expression. Last, Nakazawa et al. (Genes Cells, 2015) found that interphase binding of condensin correlates with loci of high gene expression, and used heat shock to demonstrate how such accumulation can be redirected upon gene expression program changes (quite like what is seen by the authors for senescence genes in OIS). Critically though, polIII-mediated transcription was neither repressed nor activated by condensin, and transcription levels did not change even when mutant condensin was used that could not associate with chromosomal DNA -- again, the only effect was massive chromosome mis-segregation.

As a result, it remains a challenge to decouple the effects of condensin subunits in cell cycle progression from those potentially occurring only in interphase chromatin regulation. The authors have done a large range of analyses to dissect the potential interphase role of condensin II in proliferating and senescent cells, and have tried to reconcile a number of surprising, potentially interesting, but often contradictory findings. I acknowledge that using arrested cells to study these complexes offers an advantage, but there needs to be more work added to fully clarify four major points that are the backbone of this work:

1. To confirm that CAP-H2 binding is not only due to the highly-transcribed, highly-accessible nature of the mapped loci.
2. To definitively show that CAP-H2 represents a full condensin II complex and is not moonlighting there as part of a different one.
3. To reconcile opposing roles for the protein complex that might very well arise by having a fraction of still-dividing cells in the study population.
4. To give details on experiments and analyses that are insofar missing.

Additional more specific points that also need to be addressed are as follows.

Introduction:

- For the sake of completion, this section needs to refer to the recent Zirkel et al, Mol Cell, 2018 paper on replicative senescence entry-induced 3D genome reorganization. This particular paper also claims much less A-to-B/B-to-A transitions early in replicative senescence compared to the “deep” senescence Hi-C data the authors refer to here (from Criscione et al, 2015).
- Also, I think the authors should be more cautious about the McCord et al., Genome Res, 2012 claims, as that manuscript contains two progeria-related Hi-C datasets at two different passages and only late-passage progeroid cells show the mentioned effect.
- Last, all the contradictory pieces of evidence that I list above used be presented and discussed in context in this paper's introduction.

Results:

- Fig. 1: again, the correlation of condensin bound to active genes could be attributed simply to their high transcript production (see comments above); also, the TF co-associations the authors claim are not supported by ChIP data but rather by motif analyses. I would suggest to improve this by adding some ChIP-seq data and/or by using a more sensitive analysis approach for motifs (e.g., HINT by Gusmao et al., Nat Methods, 2015, where only motifs called accessible using Dnase footprinting are scanned).
- Fig. 2 and general Hi-C comments: I could not find anywhere the sequencing depth the authors used for their Hi-C libraries (or for the ChIP-/RNA-seq libraries either, for that matter), and also there is no mention of the per cent of mapped and duplicated reads, the way that optimal Hi-C analysis resolution was estimated, or whether re-performing analyses at different resolutions changes the findings of the authors. Moreover, it would be nice to see how their data compare to the findings of the first OIS Hi-C data by Chandra et al. (although it seems that only a low-res such comparison is indeed feasible).
- SAHFs: the claim that the long-range heterochromatin interactions likely represent SAHF formation, must be backed up by 3D-DNA FISH experiments. Otherwise it is pure presumption. Moreover, the authors state that “SAHF organization is functionally linked to the down-regulation of genes located within the 400-700 kb range”; this would imply some positional periodicity/clustering of senescence-suppressed genes? To date, not such “deterministic” periodicity has been recorded in the genomes of higher eukaryotes (aside from small clusters like those of histone genes). The authors must address this, or remove the claim altogether.
- A-to-B/B-to-A transitions: despite giving the exact same P-value (of <0.001) the authors claim that B-to-A transitions are most conserved between their two replicates and A-to-B are “somewhat conserved”. How can this be? Also, do these exact transitions appear in the Chandra et al. data? The authors themselves also acknowledge that the prevalence of the B-to-A transition was unexpected, but it is not clear to me how this is justified by the gene expression, chromatin marking and physiological changes OIS cells undergo. Also, looking into the replicative senescence examples, those genomes produce less nascent RNA overall (~15% less; Zirkel et al., 2018) and only show upregulation or repetitive sequences (Criscione et al., 2015). Can these be reconciled? Especially since the authors state that “relative ChIP-seq enrichment of epigenetic marks was not drastically altered between OIS and growing cells, for any compartmental categories”? It would also be worthwhile for the authors to perform some more quantitative assays – e.g., they could use ELISA-based methods to see overall changes in particular histone marks (shown to change in RS at least), and/or pulse label nascent RNA with A488-EUTP and quantify using microscopy to see if there is global elevation or drop in transcription in their OIS model.
- The statement “Alternatively, it is also possible that SAHF organization promotes BA transitions upon OIS” contrasts the authors' finding it seems; what model would allow for this and which of the data in this manuscript are in support of this model? Generally, it seems imperative that the authors use

microscopy to model domain decompaction in their B-to-A transitioning regions, also in relation to the SAHF-forming compartments they map (where microscopy will also be needed).

- TADs: Unless I am mistaken, the fraction/number of TADs changing between OIS and growing cells is in the order of 7-9%, but the effect of the condensin-KD on TADs was not assessed. Also, the exemplary maps in Fig. 3h are rather noisy, again begging the question: how was the highest and optimal resolution assessed? Moreover, in the neighbouring panel Fig 3e, the CAP-H2 ChIP-seq data is of high background and with local peaks that do not exceed 7 rpm; this is worrisome when correlated to TAD/domain boundaries at over-25-kbp resolution – and also leads to my next point.

- CAP-H2 ChIP-seq: the authors tried quite hard to characterise the antisera used for their ChIP, and although not as efficient as the FLAG-mediated pulldown, it looks comparable. Nonetheless, throughout the manuscript we only see 4 browser views of data. In these the read enrichments are quite low and the background not negligible when using the Bethyl antibody. Thus, I am guessing the peak calling was severely affected – but there are not specific details on this, nor are there any tracks showing input samples from OIS and growing cells. In Suppl. Fig. 1a there is a mention of 66,500 CAP-H2 peaks, whereas in Suppl. Fig. 1e these are <7,000, and to add to the confusion in Suppl. Fig. 1d the ChIP-seq signal is now presented over 2-kbp genomic bins without clarifying any threshold (or the exact benefit of such an approach). Then, for growing cells at least, there should be an estimation of the fraction of cells in each cell cycle phase, so as to even indirectly show that negligible (or not) ChIP-seq signal comes from mitotic cells. Most importantly though, the browser views in Fig. 1a are in agreement with the work I mentioned above (Nakazawa et al., 2015), whereby strong transcription seems to drive condensin accumulation. The profiles seen along these three loci, as well as the metagene profiles in the supplement, point to binding peaks covering the whole gene body and to strong correlation with polII occupancy and transcription. This makes it hard to assess the ChIP-seq quality, since it is now well accepted that many of the spurious and non-specific peaks one obtains in ChIP-seq experiments result from highly-transcribed loci (e.g., Teytelman et al., PNAS, 2013). It is imperative to address all these points as the ChIP data are central to the observations that build this work.

- CAP-H2 knockdown: Performing these KDs in OIS cells is a nice means of trying to decouple the cell cycle from the interphase role of this complex. Nonetheless, if I understand correctly, the plots for BrdU and SA- β -gal indicate that there exists a fraction of non-senesced cells in the OIS population and I am wondering whether many of the effects seen can be attributed to this. For example, it is conceivable that TP53 target genes are suppressed because in a considerable fraction of still-dividing cells the cell cycle is deregulated. Could the authors quantify the effect and clarify this?

- CAP-H2 KD Hi-C: Again, as in every other such dataset here, I am missing details on seq depth, obtained resolution, etc. Also, the statement “these combined data suggest that condensin facilitates > 1Mb long-range contacts and also plays an inhibitory role in < 100 kb local contacts” is not further explained. Which are the two mechanisms that explain such distinct modes of action for condensin II? Later on the authors suggest that “condensin - binding to genes in A compartments participates not only in genomic contacts but also in the upregulation of genes” before stating also that “condensin plays an inhibitory role in enhancer-enhancer and enhancer-promoter contacts”. This is again a direct contradiction – how can the authors resolve this? The concluding experiments, where fractionation shows condensin in the chromatin and the insoluble fractions, only add to the confusion and the microscopy attached to this will only be of use if repeated on a super-resolution platform (like a STED or OMX).

- CAP-H2 at enhancers: although there is not browser view of enhancer regions carrying condensin II, this is a finding that might explain the regulatory capacity of the complex. However, we know understand that K27ac does not suffice to characterize an enhancer as truly active and regulatory of any given gene. This would require demonstration of the spatial interaction with a promoter (which the authors could do for super-enhancers and promoter even at 40-kbp resolution), and then the functional dissection (by CRISPR that deletes the binding site inferred in the CAP-H2 ChIP for this particular enhancer and then test gene regulation by RT-qPCR). Otherwise, this enhancer part is by far

the weakest point in the manuscript and is best left out.

- Role in RS: OIS is a widely-used, but admittedly artificial senescence model (e.g., SAHFs have never been observed in vivo). Thus, the results and data obtained using RS are probably more relevant to the in vivo reality of senescence physiology and regulation. However, it seems that therein the results and transitions are less easily explained. I would thus urge the authors to rewrite the discussion and make a clear and balanced summary of the RS findings and how they might differ to the OIS ones.

Reviewer #2 (Remarks to the Author):

The manuscript by Iwasaki et al investigates the role of condensin II in 3D genome organization and transcriptional regulation in senescent cells. The authors build on their earlier work on a role for condensin II in senescence. They argue that condensin II is enriched at, and is required for the expression of certain senescence-associated genes. The authors also suggest that a common feature between different types of senescence is an increase of A compartmentalization, and that this depends on condensin II.

While the topic is interesting, the experiments performed in this manuscript to my opinion remain descriptive. The authors conclude and state in the abstract that condensin II contributes to the establishment and 'maintenance' of the senescent state. This important conclusion however is not supported by the data. The authors clearly show in fig 5D that condensin II depletion does not promote cell cycle re-entry. Sure, the expression of certain markers is affected, but these apparently are merely correlative markers that do not affect the arrested state. This then leaves the authors' observation that condensin II regulates the expression of factors that are not causal to the senescence maintenance phenotype. In the absence of any biological phenotype, I do not consider this to be a strong candidate for Nature Communications.

Further points:

1) In Extended data 1D, the authors use two different antibodies that both appear to recognize CAPH2, yet the bands they detect in senescent cells are totally different. This is a bit worrying, as the authors use one of these antibodies for their ChIPs. Is this then the real result? Is condensin II really enriched at the senescence-associated genes?

2) The RS transitions used in Fig 3 C-F are from one of their biological replicates. Supplementary Fig 4 I-L show that only a fraction of these transitions is shared between replicates. I would have thought that these shared transitions should be used for comparison between OIS and RS.

3) The model presented in 6o is entirely unclear to me. I have no idea what the authors are trying to depict here.

Reviewers' comments:

Reviewer #1 (Remarks to the Author):

In this manuscript, Iwasaki and colleagues investigate the role of condensin II complexes in interphase cells that are either proliferating or have undergone senescence. There have been sporadic reports (only from mouse ESCs for mammalian systems) that SMC2 may bind and regulate promoters and enhancers, but also contradictory data from non-stem cell models (fibroblasts). The authors have also themselves shown that yeast condensin II contributes to genome organization, outside of its established role in the cell cycle. Thus, such an interrogation is interesting and could allow some resolution in the apparent SMC2 controversy in human cells, on top of the proposed role in oncogene-induced senescence.

However, the authors themselves have published on an accentuated role for SMCs and their gene associations in mitosis (Iwasaki et al, NAR, 2016) and recent published work points to the vast majority of SMC-depletion effects stemming from chromosome mis-segregation effects, rather than from direct gene regulation events -- crucially, preventing karyotype abnormalities in daughter cells restores a normal transcriptome despite condensin inactivation (Hocquet et al, eLife, 2018). All this data is of course in yeast, so it might of course be that in higher metazoans condensin II complexes have indeed acquired additional roles during interphase. Still, in *Drosophila*, the Cap-H2 subunit works with Mrg factors to exert some gene expression control -- and according to my understanding it has been formally shown that this is a complex involving the full condensin II complex. And in *C. elegans*, work from the Dekker and Meyer labs, has shown that condensin II is essentially only important for X-chromosome functions.

Interestingly, Haeuseler et al. (Genes Dev., 2008) have shown that condensin interacts with polIII-specific factors to allow their functional nucleolar clustering in yeast cells, and depletion of essentially any subunit leads to declustering and to a loss of their insulatory function for polIII genes. Then, Sutani et al. (Nat Commun., 2015) showed how condensin II binding to active promoters is a means to prime the genome for mitosis, rather than to regulate gene expression. Last, Nakazawa et al. (Genes Cells, 2015) found that interphase binding of condensin correlates with loci of high gene expression, and used heat shock to demonstrate how such accumulation can be redirected upon gene expression program changes (quite like what is seen by the authors for senescence genes in OIS). Critically though, polIII-mediated transcription was neither repressed nor activated by condensin, and transcription levels did not change even when mutant condensin was used that could not associate with chromosomal DNA -- again, the only effect was massive chromosome mis-segregation.

As a result, it remains a challenge to decouple the effects of condensin subunits in cell cycle progression from those potentially occurring only in interphase chromatin regulation. The authors have done a large range of analyses to dissect the potential interphase role of condensin II in proliferating and senescent cells, and have tried to reconcile a number of surprising, potentially interesting, but often contradictory findings. I acknowledge that using arrested cells to study these complexes offers an advantage, but there needs to be more work added to fully clarify four major points that are the backbone of this work:

We thank the reviewer for very supportive and insightful comments. As you can see below, we have performed each experiment requested by the reviewer, and clarified areas that were potentially confusing.

1. To confirm that CAP-H2 binding is not only due to the highly-transcribed, highly-accessible nature of the mapped loci.

We performed CAP-H2 ChIP-seq using IMR90 OIS cells treated by RNA Polymerase II inhibitors (Triptolide and α -Amanitin), which allowed us to determine the relationship between CAP-H2 binding and transcription (New Supplementary Fig. 1k-o). Furthermore, we carried out nascent RNA visualization, RT-qPCR, and RNA-seq experiments and confirmed that transcription was severely impaired by Pol II inhibitor treatment. Importantly, however, CAP-H2 enrichment was retained at gene regions after the inhibitor treatment, indicating that CAP-H2 enrichment at gene

regions is not derived from spurious, non-specific ChIP-seq peaks. These new data are included in the revised manuscript.

Moreover, as shown in Fig. 5i, many genes ($n = 4,073$) up-regulated upon OIS were bound by CAP-H2 and down-regulated by CAP-H2 KD, suggesting that CAP-H2 plays a direct role in the up-regulation of senescence genes. If CAP-H2 simply binds to highly transcribed genes due to the chromatin accessibility, CAP-H2 KD is unlikely to down-regulate senescence genes.

2. To definitively show that CAP-H2 represents a full condensin II complex and is not moonlighting there as part of a different one.

We carried out additional ChIP-seq experiments for an SMC2 condensin subunit (Supplementary Fig. 1a,b). The new data suggest that these condensin subunits interact and bind to their target loci as the condensin II complex.

3. To reconcile opposing roles for the protein complex that might very well arise by having a fraction of still-dividing cells in the study population.

We performed IF microscopic experiments to visualize mitotic spindles and observed that OIS and CAP-H2 KD cells did not show spindle staining, suggesting that CAP-H2 KD does not promote cell cycle re-entry and mitotic defects (New Supplementary Fig. 5a).

It is well-established that senescence-associated growth arrest is mediated by activation of both the p53/p21 and the p16/RB pathways. We demonstrated that the transcriptional regulation of senescence-associated genes (p53 targets and SASP genes) is mediated by condensin II (Fig. 5e,f; Extended Data 3). This was further validated in the revised manuscript. Our results clearly show that the optimal expression of p53 target genes such as p21 and SASP genes such as IL1B is CAP-H2-dependent at both transcription and protein levels (Fig. 5e,f; Extended Data 3; New Fig. 5k; New Supplementary Fig. 5b). In contrast, RB phosphorylation and E2F target genes such as Cyclin A were not affected by CAP-H2 knockdown (New Supplementary Fig. 5b; Fig. 5g; Extended Data 3). These results suggest that CAP-H2 KD impairs the p53/p21 pathway but does not compromise the p16/RB pathway which inhibits E2F activation. Therefore, our results support a model whereby CAP-H2 does not participate in the p16/RB pathway, which accounts for the observed maintenance of growth arrest in CAP-H2 KD cells. Together, this study shows that CAP-H2 is necessary for the full activation of SASP and the p53/p21 pathway and that the CAP-H2 KD impairs senescence, whereas CAP-H2 KD does not affect the p16/RB pathway, which maintains the growth arrest observed in established senescent cells with CAP-H2 KD.

Please note that previous yeast work has shown that chromosomal mis-segregation caused by condensin inactivation results in elevated accumulation of gene transcripts, whereas in this study CAP-H2 condensin KD down-regulates senescence genes. Therefore, it is likely that human condensin plays an activation role in gene transcription unlike the repressive role observed for yeast condensin.

4. To give details on experiments and analyses that are insofar missing.

We provided experimental details including how in situ Hi-C data were processed by the filters (Supplementary Methods) and how the highest map resolutions were defined (Supplementary Table 1). The ChIP-seq and RNA-seq procedures are also detailed in the revised manuscript.

Additional more specific points that also need to be addressed are as follows.

Introduction:

5. For the sake of completion, this section needs to refer to the recent Zirkel et al, Mol Cell, 2018 paper on replicative senescence entry-induced 3D genome reorganization. This particular paper also claims much less A-to-B/B-to-A transitions early in replicative senescence compared to the “deep” senescence Hi-C data the authors refer to here (from Crisicione et al, 2015).

We described the paper (Zirkel et al. Mol Cell 2018) in the context of the point raised by the reviewer.

6. Also, I think the authors should be more cautious about the McCord et al., Genome Res, 2012 claims, as that manuscript contains two progeria-related Hi-C datasets at two different passages and only late-passage progeroid cells show the mentioned effect.

As suggested by the reviewer, we have revised the manuscript.

7. Last, all the contradictory pieces of evidence that I list above used be presented and discussed in context in this paper's introduction.

As recommended by the reviewer, we have discussed the indicated papers in the revised manuscript (INTRODUCTION).

Results:

8. Fig. 1: again, the correlation of condensin bound to active genes could be attributed simply to their high transcript production (see comments above); also, the TF co-associations the authors claim are not supported by ChIP data but rather by motif analyses. I would suggest to improve this by adding some ChIP-seq data and/or by using a more sensitive analysis approach for motifs (e.g., HINT by Gusmao et al., Nat Methods, 2015, where only motifs called accessible using Dnase footprinting are scanned).

As suggested by the reviewer, we performed additional ChIP-seq analysis for an SMC2 condensin subunit and confirmed that SMC2 co-localizes with CAP-H2 in OIS cells (Supplementary Fig. 1a,b).

In addition, we performed CAP-H2 ChIP-seq using IMR90 OIS cells treated by RNA Polymerase II inhibitors (Triptolide and α -Amanitin), which allowed us to determine the relationship between CAP-H2 binding and transcription (New Supplementary Fig. 1k-o). Moreover, we carried out nascent RNA visualization, RT-qPCR, and RNA-seq experiments and confirmed that transcription was impaired by Pol II inhibitor treatment. However, importantly, we observed that CAP-H2 enrichment was retained at gene regions after the inhibitor treatment, indicating that CAP-H2 enrichment at genes is not derived from spurious, non-specific ChIP-seq peaks. These new data are included in the revised manuscript.

9. Fig. 2 and general Hi-C comments: I could not find anywhere the sequencing depth the authors used for their Hi-C libraries (or for the ChIP-/RNA-seq libraries either, for that matter), and also there is no mention of the per cent of mapped and duplicated reads, the way that optimal Hi-C analysis resolution was estimated, or whether re-performing analyses at different resolutions changes the findings of the authors. Moreover, it would be nice to see how their data compare to the findings of the first OIS Hi-C data by Chandra et al. (although it seems that only a low-res such comparison is indeed feasible).

We have provided experimental details including how in situ Hi-C data were processed by several filters (Supplementary Methods) and how the highest map resolutions were defined (Supplementary Table 1). The ChIP-seq and RNA-seq procedures are also detailed in the manuscript.

Furthermore, as suggested by the reviewer, we compared our in situ Hi-C data to the WI-38 Hi-C data (Chandra et al. 2015) and observed that frequent BA transitions upon OIS was also conserved in WI-38 cells, and that BA and AB transitions upon OIS observed in IMR90 cells were significantly overlapped with those in WI-38 cells, suggesting that compartmental transitions upon OIS were at least partly conserved in different cell lines (New Supplementary Fig. 2e,f).

10. SAHFs: the claim that the long-range heterochromatin interactions likely represent SAHF formation, must be backed up by 3D-DNA FISH experiments. Otherwise it is pure presumption. Moreover, the authors state that "SAHF organization is functionally linked to the down-regulation of genes located within the 400-700 kb range"; this would imply some positional periodicity/clustering of senescence-suppressed genes? To date, not such "deterministic" periodicity has been recorded in the genomes of higher eukaryotes (aside from small clusters like those of histone genes). The authors must address this, or remove the claim altogether.

As recommended by the reviewer, we performed FISH experiments to visualize the heterochromatic (P1 and P2 probes) and BA-switching loci (P3 and P4 probes) in OIS cells and observed that the P1 and P2 probes were positioned significantly closer to the center of SAHF compared to the P3 and P4 probes ($P = 1.69 \times 10^{-12}$, two-sided Mann–Whitney U test), indicating that heterochromatic loci are typically embedded within SAHF. This important new result has been included in the revised manuscript (New Fig. 2h; Supplementary Fig. 2h,i).

Moreover, we observed that genes located within the 400-700 kb range from SAHF were significantly enriched with down-regulated genes (Fig. 2i-l). In addition, genes located within the 500-700 kb range from SAHF were more frequently down-regulated compared to the control considering all human genes ($P = 0.00641$, two-sided Mann–Whitney U test; Fig. 2m). That being said, the slight down-regulation of genes located in the 400-500 kb range occasionally occurs for a limited number of genes. Therefore, we revised the manuscript to describe that SAHF organization potentially has a weak position effect on expression of genes in proximity, which was observed in this study.

11. A-to-B/B-to-A transitions: despite giving the exact same P-value (of <0.001) the authors claim that B-to-A transitions are most conserved between their two replicates and A-to-B are “somewhat conserved”. How can this be? Also, do these exact transitions appear in the Chandra et al. data? The authors themselves also acknowledge that the prevalence of the B-to-A transition was unexpected, but it is not clear to me how this is justified by the gene expression, chromatin marking and physiological changes OIS cells undergo. Also, looking into the replicative senescence examples, those genomes produce less nascent RNA overall (~15% less; Zirkel et al., 2018) and only show upregulation or repetitive sequences (Criscione et al., 2015). Can these be reconciled? Especially since the authors state that “relative ChIP-seq enrichment of epigenetic marks was not drastically altered between OIS and growing cells, for any compartmental categories”? It would also be worthwhile for the authors to perform some more quantitative assays – e.g., they could use ELISA-based methods to see overall changes in particular histone marks (shown to change in RS at least), and/or pulse label nascent RNA with A488-EUTP and quantify using microscopy to see if there is global elevation or drop in transcription in their OIS model.

To test for common reorganizations, we compared regions that undergo BA and AB transitions upon OIS and upon RS. Indeed, we observed significant overlap of BA transitions upon OIS and RS in IMR90 cells ($P = 2.71 \times 10^{-973}$, hypergeometric distribution test; Fig. 4c) and also significant overlap of AB transitions upon OIS and RS to a lesser extent ($P = 6.01 \times 10^{-86}$, hypergeometric distribution test; Supplementary Fig. 4m). We provided the p values in the revised manuscript.

Furthermore, we observed that frequent BA transitions upon OIS were also conserved in WI-38 cells (Chandra et al., 2015), and that BA and AB transitions upon OIS observed in IMR90 cells were significantly overlapped with those in WI-38 cells, indicating that compartmental transitions upon OIS were at least partly conserved in different cell lines (new Supplementary Fig. 2e,f).

As recommended by the reviewer, we visualized nascent RNA in OIS and growing cells. Cells were subsequently incubated with 5-ethynyl uridine (EU) for 1 hour. EU was incorporated in nascent RNA and reacted with Cy3-azide via click chemistry. This analysis was detailed in Supplementary Methods. We observed that global expression levels were slightly higher in OIS cells than in growing cells ($P = 4.9 \times 10^{-9}$, two-sided Mann–Whitney U test; Supplementary Fig. 1k), suggesting that many genes are actively transcribed in OIS cells as in growing cells (New Supplementary Fig. 1k). In addition, it has been shown that senescent processes involve radical changes in gene expression, where especially genes encoding the senescence-associated secretory phenotype (SASP) factors are highly activated in senescent cells.

12. The statement “Alternatively, it is also possible that SAHF organization promotes BA transitions upon OIS” contrasts the authors’ finding it seems; what model would allow for this and which of the data in this manuscript are in support of this model? Generally, it seems imperative that the authors use microscopy

to model domain decompaction in their B-to-A transitioning regions, also in relation to the SAHF-forming compartments they map (where microscopy will also be needed).

The reviewer's point is well-taken. We removed the indicated sentence in the revised manuscript.

13. TADs: Unless I am mistaken, the fraction/number of TADs changing between OIS and growing cells is in the order of 7-9%, but the effect of the condensin-KD on TADs was not assessed. Also, the exemplary maps in Fig. 3h are rather noisy, again begging the question: how was the highest and optimal resolution assessed? Moreover, in the neighbouring panel Fig 3e, the CAP-H2 ChIP-seq data is of high background and with local peaks that do not exceed 7 rpm; this is worrisome when correlated to TAD/domain boundaries at over-25-kbp resolution – and also leads to my next point.

As suggested by the reviewer, we examined how CAP-H2 KD affects TAD borders and observed that sizes and numbers of TADs in CAP-H2 KD cells were similar to those in OIS cells (Supplementary Fig. 3d) and that 7-9% TAD borders were specific to either OIS or CAP-H2 KD cells (new Supplementary Fig.5I).

In the revised manuscript, we have described how the highest map resolution was defined as previously described (Durand et al. 2016; Supplementary Table 1). The map resolution in Fig. 3h is 40 kb, which is lower than the highest resolution recommend by the literature. TAD and compartmental reorganizations upon OIS were monitored by border strength scores and PCA scores, respectively.

14. CAP-H2 ChIP-seq: the authors tried quite hard to characterise the antisera used for their ChIP, and although not as efficient as the FLAG-mediated pulldown, it looks comparable. Nonetheless, throughout the manuscript we only see 4 browser views of data. In these the read enrichments are quite low and the background not negligible when using the Bethyl antibody. Thus, I am guessing the peak calling was severely affected – but there are not specific details on this, nor are there any tracks showing input samples from OIS and growing cells. In Suppl. Fig. 1a there is a mention of 66,500 CAP-H2 peaks, whereas in Suppl. Fig. 1e these are <7,000, and to add to the confusion in Suppl. Fig. 1d the ChIP-seq signal is now presented over 2-kbp genomic bins without clarifying any threshold (or the exact benefit of such an approach). Then, for growing cells at least, there should be an estimation of the fraction of cells in each cell cycle phase, so as to even indirectly show that negligible (or not) ChIP-seq signal comes from mitotic cells. Most importantly though, the browser views in Fig. 1a are in agreement with the work I mentioned above (Nakazawa et al., 2015), whereby strong transcription seems to drive condensin accumulation. The profiles seen along these three loci, as well as the metagene profiles in the supplement, point to binding peaks covering the whole gene body and to strong correlation with polII occupancy and transcription. This makes it hard to assess the ChIP-seq quality, since it is now well accepted that many of the spurious and non-specific peaks one obtains in ChIP-seq experiments result from highly-transcribed loci (e.g., Teytelman et al., PNAS, 2013). It is imperative to address all these points as the ChIP data are central to the observations that build this work.

In the revised manuscript, we described in details regarding how CAP-H2 peaks were defined (Supplementary Methods). The input DNA control sample from the ChIP procedure was subjected to the same bioinformatic analysis and employed for peak calling; Genomic sections carrying relative enrichment scores between ChIP and Input DNA samples above 2-fold, $P < 0.0001$ and FDR < 0.05 were defined as peaks. Furthermore, CAP-H2 binding sites were predicted as common sites shared by multiple ChIP-seq data (Supplementary Fig. 1c; Supplementary Notes). Moreover, to avoid any confusion, the numbers of CAP-H2 binding sites predicted for OIS and growing cells are consistent across the revised manuscript (New Supplementary Fig. 1).

We also performed CAP-H2 ChIP-seq using IMR90 OIS cells treated by RNA Polymerase II inhibitors (Triptolide and α -Amanitin), which allowed us to determine the relationship between CAP-H2 binding and transcription (Supplementary Fig. 1k-o). Furthermore, we carried out in situ RNA visualization, RT-qPCR, and RNA-seq experiments and confirmed that transcription was impaired by Pol II inhibitor treatment. Importantly, however, CAP-H2 enrichment was retained at gene regions after the inhibitor treatment, indicating that CAP-H2 enrichment at gene regions is not

derived from spurious, non-specific ChIP-seq peaks. These new data are included in the revised manuscript.

Moreover, as shown in Fig. 5i, many genes (n = 4,073) up-regulated upon OIS were bound by CAP-H2 and down-regulated by CAP-H2 KD, suggesting that CAP-H2 plays a direct role in the up-regulation of senescence genes. If CAP-H2 simply binds to highly transcribed genes due to the chromatin accessibility, CAP-H2 KD is unlikely to down-regulate senescence genes.

15. CAP-H2 knockdown: Performing these KDs in OIS cells is a nice means of trying to decouple the cell cycle from the interphase role of this complex. Nonetheless, if I understand correctly, the plots for BrdU and SA- β -gal indicate that there exists a fraction of non-senesced cells in the OIS population and I am wondering whether many of the effects seen can be attributed to this. For example, it is conceivable that TP53 target genes are suppressed because in a considerable fraction of still-dividing cells the cell cycle is deregulated. Could the authors quantify the effect and clarify this?

We performed IF microscopic experiments to visualize mitotic spindles and observed that OIS and CAP-H2 KD cells did not show spindle staining, strongly suggesting that CAP-H2 KD neither promotes cell cycle re-entry nor causes mitotic defects in OIS cells (Supplementary Fig. 5a).

It is well-established that senescence-associated growth arrest is mediated by activation of both the p53/p21 and the p16/RB pathways. Here, we demonstrated that the transcriptional regulation of senescence-associated genes (p53 targets and SASP genes) is mediated by condensin II (Fig. 5e,f; Extended Data 3). This was further validated in the revised manuscript. Our results clearly show that the optimal expression of p53 target genes such as p21 and SASP genes such as IL1B is CAP-H2-dependent at both transcription and protein levels (Fig. 5e,f; Extended Data 3; New Fig. 5k; New Supplementary Fig. 5b). In contrast, RB phosphorylation and E2F target genes such as Cyclin A were not affected by CAP-H2 knockdown (New Supplementary Fig. 5b; Fig. 5g; Extended Data 3). These results suggest that CAP-H2 KD impairs the p53/p21 pathway but does not compromise the p16/RB pathway which inhibits E2F activation. Therefore, our results support a model whereby CAP-H2 does not participate in the p16/RB pathway, which accounts for the observed maintenance of growth arrest in CAP-H2 KD cells. Together, this study shows that CAP-H2 is necessary for the full activation of SASP and the p53/p21 pathway and that CAP-H2 KD impairs senescence, whereas CAP-H2 KD does not affect the p16/RB pathway, which maintains the growth arrest observed in established senescent cells with CAP-H2 KD.

Please note that previous yeast work has shown that chromosomal mis-segregation caused by condensin inactivation results in elevated accumulation of gene transcripts, whereas in this study CAP-H2 condensin KD down-regulates senescence genes. Therefore, it is likely that human condensin plays an activation role in gene transcription unlike the repressive role observed for yeast condensin.

16. CAP-H2 KD Hi-C: Again, as in every other such dataset here, I am missing details on seq depth, obtained resolution, etc. Also, the statement “these combined data suggest that condensin facilitates > 1Mb long-range contacts and also plays an inhibitory role in < 100 kb local contacts” is not further explained. Which are the two mechanisms that explain such distinct modes of action for condensin II? Later on the authors suggest that “condensin - binding to genes in A compartments participates not only in genomic contacts but also in the upregulation of genes” before stating also that “condensin plays an inhibitory role in enhancer-enhancer and enhancer-promoter contacts”. This is again a direct contradiction – how can the authors resolve this? The concluding experiments, where fractionation shows condensin in the chromatin and the insoluble fractions, only add to the confusion and the microscopy attached to this will only be of use if repeated on a super-resolution platform (like a STED or OMX).

In the revised manuscript, we described how the highest map resolution was defined for the CAP-H2 KD data (Durand et al. 2016; Supplementary Table 1). We removed the indicated statement “these combined data suggest that condensin facilitates > 1Mb long-range contacts and also plays an inhibitory role in < 100 kb local contacts.” We also removed the sentence, “condensin plays an

inhibitory role in enhancer-enhancer and enhancer-promoter contacts". These associated sentences are removed, because the enhancer-related part (previous Supplementary Fig. 6) was discarded as suggested by the reviewer below.

As suggested by the reviewer, we carried out super-resolution microscopy (Zeiss LSM 880 with Airyscan) to visualize CAP-H2 localization in OIS cells and observed that the CAP-H2 localization pattern represented a mixture of speckled and potential filamentous distributions. In the revised manuscript, the results predicting the potential insoluble architecture are presented in the Supplement Information (Supplemental Fig. 6), and a possible mechanism involving the architecture is briefly discussed.

17. CAP-H2 at enhancers: although there is not browser view of enhancer regions carrying condensin II, this is a finding that might explain the regulatory capacity of the complex. However, we know understand that K27ac does not suffice to characterize an enhancer as truly active and regulatory of any given gene. This would require demonstration of the spatial interaction with a promoter (which the authors could do for super-enhancers and promoter even at 40-kbp resolution), and then the functional dissection (by CRISPR that deletes the binding site inferred in the CAP-H2 ChIP for this particular enhancer and then test gene regulation by RT-qPCR). Otherwise, this enhancer part is by far the weakest point in the manuscript and is best left out.

As recommended by the reviewer, we removed the previous Supplementary Fig. 6 showing the potential role of CAP-H2 binding to enhancers in gene contacts and transcriptional regulation.

18. Role in RS: OIS is a widely-used, but admittedly artificial senescence model (e.g., SAHFs have never been observed in vivo). Thus, the results and data obtained using RS are probably more relevant to the in vivo reality of senescence physiology and regulation. However, it seems that therein the results and transitions are less easily explained. I would thus urge the authors to rewrite the discussion and make a clear and balanced summary of the RS findings and how they might differ to the OIS ones.

As suggested by the reviewer, we revised the manuscript to describe that SAHF formation upon OIS is not a conserved senescent phenomenon nor has it been observed in endogenous tissues. In the revised manuscript, we described that BA-switching regions upon OIS significantly overlap with those upon RS, demonstrating that BA transitions are at least partly conserved between the two types of senescence. We also mentioned that the condensin-localized architecture observed in OIS cells might not be formed in RS cells.

Reviewer #2 (Remarks to the Author):

The manuscript by Iwasaki et al investigates the role of condensin II in 3D genome organization and transcriptional regulation in senescent cells. The authors build on their earlier work on a role for condensin II in senescence. They argue that condensin II is enriched at, and is required for the expression of certain senescence-associated genes. The authors also suggest that a common feature between different types of senescence is an increase of A compartmentalization, and that this depends on condensin II.

While the topic is interesting, the experiments performed in this manuscript to my opinion remain descriptive. The authors conclude and state in the abstract that condensin II contributes to the establishment and 'maintenance' of the senescent state. This important conclusion however is not supported by the data. The authors clearly show in fig 5D that condensin II depletion does not promote cell cycle re-entry. Sure, the expression of certain markers is affected, but these apparently are merely correlative markers that do not affect the arrested state. This then leaves the authors' observation that condensin II regulates the expression of factors that are not causal to the senescence maintenance phenotype. In the absence of any biological phenotype, I do not consider this to be a strong candidate for Nature Communications.

We thank the reviewer for supportive and insightful comments. As you can see below, we have performed each experiment requested by the reviewer, and clarified areas that were potentially confusing.

It is well-established that senescence-associated growth arrest is mediated by activation of both the p53/p21 and the p16/RB pathways. We demonstrated that the transcriptional regulation of senescence-associated genes (p53 targets and SASP genes) is mediated by condensin II (Fig. 5e,f; Extended Data 3). This was further validated in the revised manuscript. Our results clearly show that the optimal expression of p53 target genes such as p21 and SASP genes such as IL1B is condensin-dependent at both transcription and protein levels (Fig. 5e,f; Extended Data 3; New Fig. 5k; New Supplementary Fig. 5b). In contrast, RB phosphorylation and E2F target genes such as Cyclin A were not affected by condensin knockdown (New Supplementary Fig. 5b; Fig. 5g; Extended Data 3). These results reveal that condensin KD impairs the p53/p21 pathway but does not compromise the p16/RB pathway which inhibits E2F activation. Therefore, our results support a model whereby condensin does not participate in the p16/RB pathway, which accounts for the observed maintenance of growth arrest in condensin KD cells. This study shows that condensin is necessary for the full activation of SASP and the p53/p21 pathway and that condensin KD impairs senescence, whereas condensin KD does not affect the p16/RB pathway, which maintains the growth arrest observed in established senescent cells with condensin KD.

Since the p53/p21 pathway is one of the most crucial mechanisms for regulating senescence, our work revealing that condensin II contributes to activation of the p53/p21 pathway has substantial impacts on understanding the biological phenotype of cellular senescence. It is also well-recognized that the regulation of SASP factors connects senescence with aging and cancer. In this regard, our work demonstrating that condensin up-regulates SASP genes through compartmental reorganization will shed new light on a senescence phenotype related to aging and cancer.

Further points:

1. In Extended data 1D, the authors use two different antibodies that both appear to recognize CAPH2, yet the bands they detect in senescent cells are totally different. This is a bit worrying, as the authors use one of these antibodies for their ChIPs. Is this then the real result? Is condensin II really enriched at the senescence-associated genes?

We confirmed that CAP-H2 proteins were detected by the Bethyl CAP-H2 antibody and were much decreased by CAP-H2 KD (Extended Data 1b,d). The same antibody detected exogenous CAP-H2-FLAG proteins, indicating that this antibody recognizes CAP-H2 proteins (Extended Data 1e).

As suggested by the reviewer, we performed additional ChIP-seq analysis for an SMC2 condensin subunit and confirmed that SMC2 co-localizes with CAP-H2 in OIS cells, strongly suggesting that CAP-H2 binds to genomic regions as a part of the condensin II complex (Supplementary Fig. 1a,b).

2. The RS transitions used in Fig 3 C-F are from one of their biological replicates. Supplementary Fig 4 I-L show that only a fraction of these transitions is shared between replicates. I would have thought that these shared transitions should be used for comparison between OIS and RS.

As suggested by the reviewer, compartmental transitions shared by biological replicates were subjected to the analysis described in previous Figure 4c-f (new Fig. 4c,d; new Supplementary Fig. 4m,n). We observed significant overlap of BA transitions upon OIS and RS in IMR90 cells ($P = 2.71 \times 10^{-973}$, hypergeometric distribution test; Fig. 4c) and upon RS in IMR90 and BJ cells ($P = 2.74 \times 10^{-227}$, hypergeometric distribution test; Fig. 4d). We also observed significant overlap of AB transitions upon OIS and RS in IMR90 cells ($P = 6.01 \times 10^{-86}$, hypergeometric distribution test; Supplementary Fig. 4m) and RS in IMR90 and BJ cells ($P = 1.51 \times 10^{-35}$, hypergeometric distribution test; Supplementary Fig. 4n).

3. The model presented in 6o is entirely unclear to me. I have no idea what the authors are trying to depict here.

We have hopefully replaced the previous model with a simpler, easy to understand model in the revised manuscript (Fig. 6o). The model shows that the roles of condensin are to reinforce euchromatic A compartments and to promote/maintain BA transitions upon senescence, both of which are connected to the up-regulation of senescence genes.

Reviewers' comments:

Reviewer #1 (Remarks to the Author):

I the revised version of this work on condensin in OIS, Iwasaki and colleagues have indeed made a commendable effort to address all points raised by both reviewers. In addition, one must acknowledge that it remains a challenge to study complexes like this one that also have major effects on cell division, and the use of OIS is a nice way out. In its current form the manuscript is, in my opinion, significantly improved, and should be accepted for publication pending the addressing of the following (mostly minor) issues:

1. In the Venn diagram of different CAP-H2 ChIP-seq replicates there are incredible discrepancies in the number of peaks between replicates generated using the same antisera (e.g. for Bethyl-Ab ~10,000 peaks in OIS#1 versus >18,000 in OIS#2). How do the authors explain such differences?
2. In EU-labeling experiments of nascent RNA, the authors incubated IMR90 with the precursor for 1h. Surely (and this is the case in our hands, consistently) at this long time point both the nucleus and the cytoplasm will be flooded with EU-labeled steady-state mRNAs, but it does not seem case. Can the authors explain that? One would typically only incubate with EUTP for 5-15 min to get nice nascent RNA nuclear patterns. I think this ultra-long labelling scheme is what led the authors to state that, unlike in RS, OIS cells do not show reduced nascent RNA levels. This must be clarified, as can be very misleading for the field.
3. The use of triptolide and α -amanitin prior to CAP-H2 ChIP-seq is a welcome addition to the manuscript, but we need to acknowledge that it is unclear whether any of the two drugs actually change the accessibility of the bound promoters significantly. Thus, I would urge the authors to rewrite this part of the Results to explain this.
4. The western blot images shown here (and this of others in both OIS and RS) show a clear reduction of condensin levels in senescent cells. However, the authors present very nice images of senescent nuclei that show very strong IF signal for CAP-H2 (comparable to that of growing cells, see Suppl. Fig 6a,b). How do the authors explain this? Can they please quantify these fluorescent levels across ~100 cells and present a box plot alongside the images so that the readers get the full impression of their microscopy data?

Finally, I am simply offering a note of precaution: using the Durand et al analysis pipeline (as I understand from Suppl. Table 1) the authors find that the effective resolution of the in situ Hi-C maps is on average 20 kbp. However, with ~140 million read pairs in their "deepest" Hi-C replicate and with ~60 in their "shallowest", and with an on average 40% of reads being useful after all filtering, I cannot understand how such sub-25-kbp resolutions are achieved. However, it is my impression that all analyses were performed on 40-kbp resolution matrices, and I would suggest this stays strictly so.

Reviewer #2 (Remarks to the Author):

The authors have made considerable efforts to address the reviewers comments. My main point of critique however still stands. The authors clearly show that condensin II is not required for the maintenance of senescence. This is a fine result, and I would suggest that the authors don't try to claim the opposite. Strangely, in the abstract, the main text, and in the figure legends the authors still state that condensin II contributes to/facilitates the establishment and maintenance of the senescent state. This makes no sense at all.

The authors should simply stick to the facts. They could say that condensin II plays a role in the establishment and not the maintenance of this state. They could for example then add that condensin II does contribute to the expression of a handful of senescence-associated markers, but that these factors evidently don't affect the maintenance of senescence.

Another remaining issue to me is the added SMC2 ChIP. The enrichment of SMC2 at senescence-associated genes is extremely weak. This to my opinion does not strengthen the authors point.

The revised model in figure 60 is a major improvement.

Reviewers' comments:

Reviewer #1:

I the revised version of this work on condensin in OIS, Iwasaki and colleagues have indeed made a commendable effort to address all points raised by both reviewers. In addition, one must acknowledge that it remains a challenge to study complexes like this one that also have major effects on cell division, and the use of OIS is a nice way out. In its current form the manuscript is, in my opinion, significantly improved, and should be accepted for publication pending the addressing of the following (mostly minor) issues:

We thank the reviewer for very supportive comments (underlined) and insightful suggestions. As you can see below, we have performed each experiment requested by the reviewer, and clarified areas that were potentially confusing.

1. In the Venn diagram of different CAP-H2 ChIP-seq replicates there are incredible discrepancies in the number of peaks between replicates generated using the same antisera (e.g. for Bethyl-Ab ~10,000 peaks in OIS#1 versus >18,000 in OIS#2). How do the authors explain such differences?

CAP-H2 ChIP-seq peaks were relatively weak compared to other factors such as RNA Polymerase II (Fig. 1a). Therefore, minor CAP-H2 peaks were not consistently detected as binding sites. To overcome this problem and address the point previously raised by the reviewer, we carefully examined CAP-H2 ChIP-seq peaks. In the revised manuscript, we have described in detail how CAP-H2 condensin peaks were defined (Supplementary Methods). The input DNA control sample from the ChIP procedure was subjected to the same bioinformatic analysis and employed for peak calling; genomic sections carrying relative enrichment scores between ChIP and Input DNA samples above 2-fold, $P < 0.0001$ and $FDR < 0.05$ were defined as peaks. Furthermore, CAP-H2 binding sites were predicted as common sites shared by multiple ChIP-seq data (Supplementary Fig. 1c; Supplementary Notes).

2. In EU-labeling experiments of nascent RNA, the authors incubated IMR90 with the precursor for 1h. Surely (and this is the case in our hands, consistently) at this long time point both the nucleus and the cytoplasm will be flooded with EU-labeled steady-state mRNAs, but it does not seem case. Can the authors explain that? One would typically only incubate with EUTP for 5-15 min to get nice nascent RNA nuclear patterns. I think this ultra-long labelling scheme is what led the authors is stating that, unlike in RS, OIS cells do not show reduced nascent RNA levels. This must be clarified, as can be very misleading for the field.

As described in a previously published paper (Jao and Salic, 2008, PNAS), human IMR90 cells were incubated with EU for 1 hour to visualize nascent RNAs. As shown in Figure 1C of the PNAS paper, EU-incorporated nascent RNA signals are continuously enhanced during the EU treatment between 30 minutes and 3 hours in NIH3T3 cells, which grow faster than IMR90 cells, suggesting that EU signals in IMR90 cells are unlikely to be saturated during the 1-hour EU treatment. Moreover, EU-Cy3 fluorescent signals in Supplementary Fig. 1k were accumulated predominantly in nuclei and diminished by treatment of the RNA polymerase inhibitor, Triptolide, again suggesting that the 1-hour EU treatment allows us to detect nascent RNAs. That being said, as suggested by the reviewer, we also incubated IMR90 growing and OIS cells with EU for 10 minutes. We again found that global gene expression levels were significantly higher in OIS cells than in growing cells (Result shown below). Since this result was the essentially same as that shown in Supplementally Fig. 1k, we did not include the redundant data in the revised manuscript. Please note that there is no additional space in Supplementary Fig. 1.

3. The use of triptolide and α -amanitin prior to CAP-H2 ChIP-seq is a welcome addition to the manuscript, but we need to acknowledge that it is unclear whether any of the two drugs actually change the accessibility of the bound promoters significantly. Thus, I would urge the authors to rewrite this part of the Results to explain this.

As suggested by the reviewer, we described that it is still unclear how the inhibitor treatment affects the protein accessibility at gene regions.

4. The western blot images shown here (and this of others in both OIS and RS) show a clear reduction of condensin levels in senescent cells. However, the authors present very nice images of senescent nuclei that show very strong IF signal for CAP-H2 (comparable to that of growing cells, see Suppl. Fig 6a,b. How do the authors explain this? Can they please quantify these fluorescent levels across ~100 cells and present a box plot alongside the images so that the readers get the full impression of their microscopy data?

CAP-H2 signals shown in the IF images (Supplementary Fig. 1a,b) are consistently derived from IMR90 OIS cells, but not from growing cells. As suggested by the reviewer, we quantified CAP-H2 signals in more than 300 OIS cells, and fluorescent signals are summarized as a boxplot below with an IF image. The manuscript has been revised to avoid any confusion and to reflect this point.

Finally, I am simply offering a note of precaution: using the Durand et al analysis pipeline (as I understand from Suppl. Table 1) the authors find that the effective resolution of the in situ Hi-C maps is on average 20 kbp. However, with ~140 million read pairs in their "deepest" Hi-C replicate and with ~60 in their "shallowest", and with an on average 40% of reads being useful after all filtering, I cannot understand how such sub-25-kbp resolutions are achieved. However, it is my impression that all analyses were performed on 40-kbp resolution matrices, and I would suggest this stays strictly so.

As suggested by the reviewer, we employed 40 kb or lower resolutions across the manuscript.

Reviewer #2:

The authors have made considerable efforts to address the reviewers comments. My main point of critique however still stands. The authors clearly show that condensin II is not required for the maintenance of senescence. This is a fine result, and I would suggest that the authors don't try to claim the opposite. Strangely, in the abstract, the main text, and in the figure legends the authors still state that condensin II contributes to/facilitates the establishment and maintenance of the senescent state. This makes no sense at all.

We thank the reviewer for supportive and insightful comments. As you described below, we have revised the manuscript according to the reviewer's comments.

The authors should simply stick to the facts. They could say that condensin II plays a role in the establishment and not the maintenance of this state. They could for example then add that condensin II does contribute to the expression of a handful of senescence-associated markers, but that these factors evidently don't affect the maintenance of senescence.

As suggested, we removed all sentences describing the involvement of condensin in the maintenance of senescence.

Another remaining issue to me is the added SMC2 ChIP. The enrichment of SMC2 at senescence-associated genes is extremely weak. This to my opinion does not strengthen the authors point.

In the revised manuscript, we have described in detail how CAP-H2 condensin peaks were defined (Supplementary Methods). The input DNA control sample from the ChIP procedure was subjected to the same bioinformatic analysis and employed for peak calling; genomic sections carrying relative enrichment scores between ChIP and Input DNA samples above 2-fold, $P < 0.0001$ and FDR < 0.05 were defined as peaks. Furthermore, CAP-H2 binding sites were predicted as common sites shared by multiple ChIP-seq data (Supplementary Fig. 1c; Supplementary Notes).

We also performed CAP-H2 ChIP-seq using IMR90 OIS cells treated by RNA Polymerase II inhibitors (Triptolide and α -Amanitin), which allowed us to determine the relationship between CAP-H2 binding and transcription (Supplementary Fig. 1k-o). Furthermore, we carried out in situ RNA visualization, RT-qPCR, and RNA-seq experiments and confirmed that transcription was impaired by Pol II inhibitor treatment. Importantly, however, CAP-H2 enrichment was retained at gene regions after the inhibitor treatment, indicating that CAP-H2 enrichment at gene regions is not derived from spurious, non-specific ChIP-seq peaks.

In addition, as shown in Fig. 5i, many genes (n = 4,073) up-regulated upon OIS were bound by CAP-H2 and down-regulated by CAP-H2 KD, suggesting that CAP-H2 plays a direct role in the up-regulation of senescence genes. If CAP-H2 simply binds to highly transcribed genes due to the chromatin accessibility, CAP-H2 KD would be unlikely to down-regulate senescence genes.

Moreover, we performed additional ChIP-seq analysis for an SMC2 condensin subunit and confirmed that SMC2 tends to co-localize with CAP-H2 (Supplementary Fig 1a). The co-localization between SMC2 and CAP-H2 at the *JUN* and *PHLDA1* gene loci has been provided (Supplementary Fig 1b). These results are included as examples to indicate that these condensin subunits are at least in part co-localized at gene loci. Based on these results, we only suggest that CAP-H2 binds to genomic regions as a part of the condensin II complex.

The revised model in figure 6o is a major improvement.

We appreciate the reviewer's supportive comment for our effort on improving the model figure.

REVIEWERS' COMMENTS:

Reviewer #1 (Remarks to the Author):

I find that the authors have been thorough in addressing all my concerns, while also toning down or explaining parts of their work that could have been unclear. I would certainly endorse this manuscript for publication in its current form.

REVIEWERS' COMMENTS:

Reviewer #1 (Remarks to the Author):

I find that the authors have been thorough in addressing all my concerns, while also toning down or explaining parts of their work that could have been unclear. I would certainly endorse this manuscript for publication in its current form.

We thank the reviewer(s) for very supportive comments and insightful suggestions during the reviewing process.